# Chronic neurotransmission increases the susceptibility of lateral-line hair cells to ototoxic insults

**Daria Lukasz[1,2], Alisha Beirl[1], Katie Kindt[1]***

[1]Section on Sensory Cell Development and Function, National Institute on Deafness and Other Communication Disorders, National Institutes of Health, Bethesda, United States; [2]Department of Biology, Johns Hopkins University, Baltimore, United States

**Abstract** Sensory hair cells receive near constant stimulation by omnipresent auditory and vestibular stimuli. To detect and encode these stimuli, hair cells require steady ATP production, which can be accompanied by a buildup of mitochondrial byproducts called reactive oxygen species (ROS). ROS buildup is thought to sensitize hair cells to ototoxic insults, including the antibiotic neomycin. Work in neurons has shown that neurotransmission is a major driver of ATP production and ROS buildup. Therefore, we tested whether neurotransmission is a significant contributor to ROS buildup in hair cells. Using genetics and pharmacology, we disrupted two key aspects of neurotransmission in zebrafish hair cells: presynaptic calcium influx and the fusion of synaptic vesicles. We find that chronic block of neurotransmission enhances hair-cell survival when challenged with the ototoxin neomycin. This reduction in ototoxin susceptibility is accompanied by reduced mitochondrial activity, likely due to a reduced ATP demand. In addition, we show that mitochondrial oxidation and ROS buildup are reduced when neurotransmission is blocked. Mechanistically, we find that it is the synaptic vesicle cycle rather than presynaptic- or mitochondrial-calcium influx that contributes most significantly to this metabolic stress. Our results comprehensively indicate that, over time, neurotransmission causes ROS buildup that increases the susceptibility of hair cells to ototoxins.

*For correspondence:
katie.kindt@nih.gov

**Competing interest:** The authors declare that no competing interests exist.

## Editor's evaluation

Lukasz and colleagues report important new results revealing how changes in zebrafish lateral line hair cell synaptic activity results in increased vulnerability to ototoxic insult. The authors provide convincing evidence for neurotransmitter release altering susceptibility to aminoglycoside exposure through experiments examining mutants where synaptic release is disrupted. Changes in synaptic activity are accompanied by modest but significant changes in mitochondrial activity, consistent with previous studies revealing that mitochondrial changes impact hair cell susceptibility to damage. This work will inform future studies on how accumulating damage contributes to hair cell damage and ultimately hearing and balance disorders.

## Introduction

Sensory hair cells are a highly active cell type present in the inner ear of all vertebrates and also uniquely present in the lateral line of aquatic vertebrates. Hair cells contain specialized synapses that work near constantly to convert sensory stimuli into electrical signals (*McPherson, 2018*). Similar to neurons, in hair cells, neurotransmission requires extensive amounts of energy (*Rangaraju et al., 2014*). These energy demands are met by ATP produced via mitochondrial respiration. Mitochondrial respiration is accompanied by the production of cytotoxic reactive oxygen species (ROS). Elevated

levels of ROS result in an accumulation of oxidative stress that renders neurons and hair cells more susceptible to insults and can ultimately lead to cell death (*Singh et al., 2019*; *Wong and Ryan, 2015*). In humans, hair-cell death or damage can result in irreversible hearing loss (*Mittal et al., 2017*). What contribution the metabolic demands of synaptic transmission have on hair-cell health or susceptibility to insults is not known.

To convert sensory stimuli into neuronal signals, hair cells rely on two critical, metabolically demanding processes: mechanosensation and synaptic transmission (*McPherson, 2018*). At the apical end of hair cells, mechanosensory bundles detect sensory stimuli (*Figure 1*). Deflection of the mechanosensory bundle opens nonspecific cation channels known as mechanoelectrical transducer (MET) channels which allows potassium and calcium into the cell. MET channel function requires a calcium gradient in the mechanosensory bundle; maintenance of this gradient requires the plasma-membrane calcium-ATPase (PMCA) and is thought to pose a large energy demand (*Dumont and Gillespie, 2000*). MET channel-dependent cationic influx depolarizes the cell and triggers synaptic transmission at the base of the hair cell. Depolarization leads to the opening of basal, voltage-gated $Ca_v1.3$ calcium channels at the hair-cell presynapse (*Brandt et al., 2003*). This calcium influx is detected by the calcium sensor Otoferlin (*Roux et al., 2006*). Otoferlin is required to couple calcium influx with exocytosis and ultimately mediate the release of glutamate-containing vesicles onto the afferent neurons (*Figure 1D*). Although the metabolic demands of mechanosensation have been implicated in increased susceptibility to ototoxic insults (*Pickett et al., 2018*), the contribution of neurotransmission to this process is unclear.

In hair cells, studies have shown that both MET channel function and presynaptic calcium influx promote mitochondrial-calcium uptake (*Pickett et al., 2018*; *Wong et al., 2019*). This link is relevant to metabolism as mitochondrial-calcium uptake can stimulate mitochondrial respiration and ATP production (*Tarasov et al., 2012*). Thus, mitochondrial-calcium uptake may serve to promote the ATP production required to maintain both MET channel function and neurotransmission. Furthermore, work in neurons has shown that the recycling and recruitment of vesicle components consumes a significant portion of the ATP needed to sustain neurotransmission (*Pulido and Ryan, 2021*). Therefore, it is possible that vesicle endo- and exo-cytosis may pose similarly high energy demands in hair cells. These energy demanding processes and the resulting ROS byproducts produced by activity-driven ATP synthesis could ultimately prove to be cytotoxic.

To study the energy demands and metabolic impact of hair-cell neurotransmission we examined hair cells in the lateral line of larval zebrafish (*Figure 1A–C*). Zebrafish allow us to study hair cells that are genetically, functionally, and morphologically similar to mammalian hair cells in vivo (*Sheets et al., 2021*). For example, in both zebrafish and mammals, $Ca_v1.3$ calcium channels and Otoferlin are essential for proper hair-cell function (*Figure 1D–D'*; *Brandt et al., 2003*; *Chatterjee et al., 2015*; *Roux et al., 2006*; *Sidi et al., 2004*). Furthermore, because zebrafish are transparent, hair cells can be easily visualized using dyes or genetically encoded indicators (GEIs) to study cellular health and integrity. GEIs can also be used for measurements of evoked activity including: mechanosensitive-, presynaptic-, or mitochondrial-calcium signals as well as exocytosis (*Lukasz and Kindt, 2018*). In addition to these tools and advantages, a large body of work has reinforced the utility of the larval zebrafish model to study hair-cell stressors, including ototoxic aminoglycosides (reviewed in: *Coffin et al., 2010*).

In this study we find that, in hair cells, chronic neurotransmission is associated with increased susceptibility to the ototoxic aminoglycoside antibiotics neomycin and gentamicin. We show that over time, neurotransmission leads to increased ROS accumulation in the cytosol and mitochondria. Importantly, we find that it is the ATP demands of the synaptic vesicle cycle rather that either presynaptic calcium influx or mitochondrial calcium uptake that leads to significant ROS buildup and ototoxin susceptibility. Understanding what processes contribute to the susceptibility of hair cells to stressors is important and may lead to the development of future therapies to prevent the loss of hearing or balance.

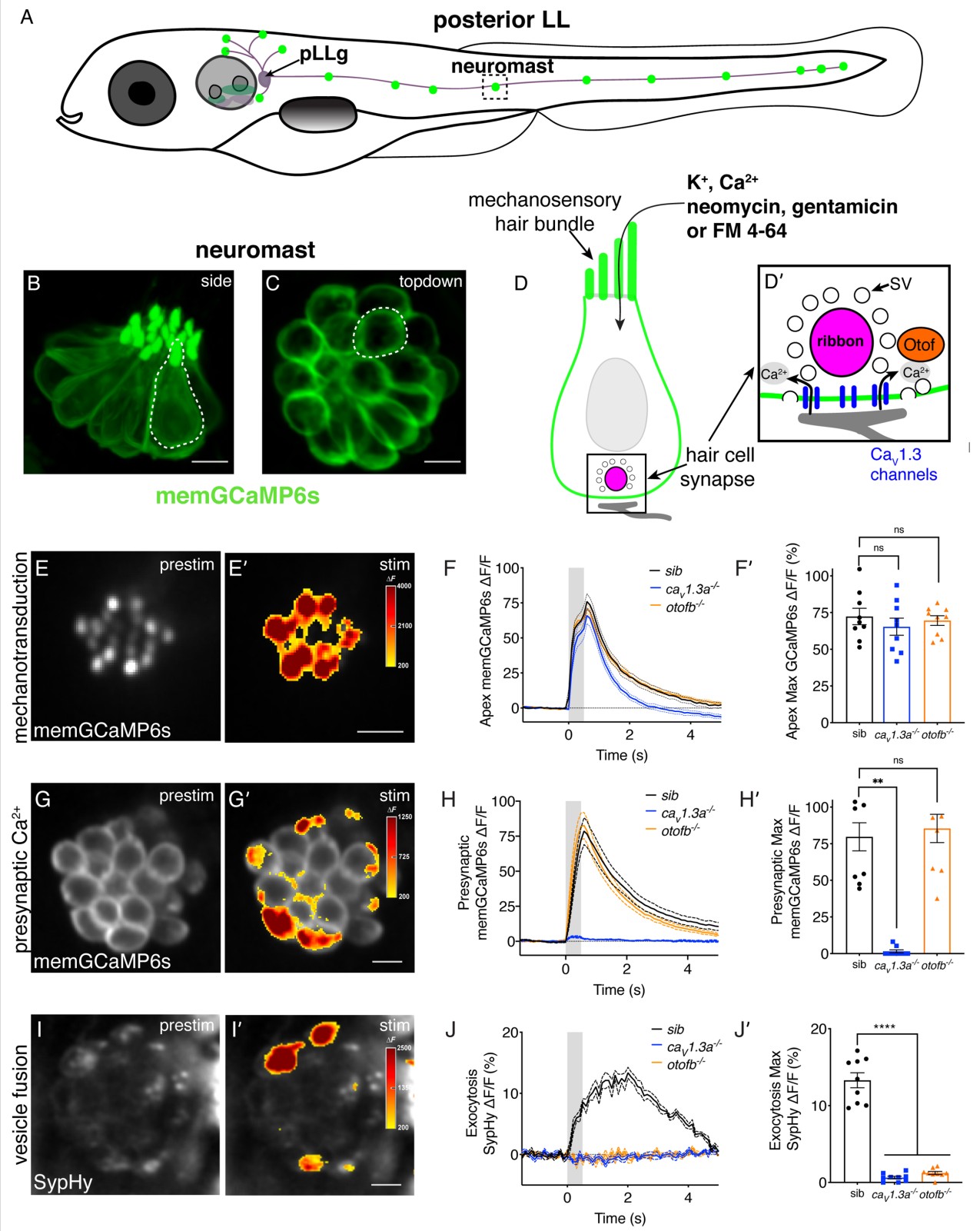

**Figure 1.** Genetic basis of neurotransmission in the zebrafish posterior-lateral line. (**A**) Cartoon drawing of 5 dpf zebrafish larva with the posterior lateral line (LL) highlighted. The LL is made up of clusters of hair cell clusters called neuromasts (green dots). Neuromasts are innervated by neurons that project from the posterior LL ganglion (pLLg). (**B–C**) Side view (**B**) and top-down view (**C**) of neuromast from *Tg[myo6b:memGCaMP6s]*^{idc1} fish where the hair-cell membrane is labeled. White dotted line demarcates a single hair cell in each image. (**D-D'**) Cartoon schematic of a side view of a hair cell.

*Figure 1 continued on next page*

*Figure 1 continued*

At the apex is the mechanosensory hair bundle. The primary pathway of entry of neomycin, gentamicin, and FM 4–64 is through mechanotransduction channels in the mechanosensory hair bundle. At the base of the hair cell is the ribbon synapse. The presynapse or ribbon (magenta) is surrounded by synaptic vesicles (SV, white circles). When mechanotransduction channels are activated, an influx of cations including calcium enters the hair bundle. Hair bundle activation leads to opening of Ca$_v$1.3 voltage-gated calcium channels (blue) and presynaptic calcium influx. The calcium sensor Otoferlin (orange) facilitates fusion by coupling calcium influx with the exocytosis of SVs and the release of glutamate onto the innervating postsynaptic afferent terminal (gray). (**E-E'**) The spatial patterns of the evoked calcium influx (GCaMP6s ΔF, indicated via the heatmaps) into sibling hair bundles (**E'**) compared to prestimulus (**C**). (**F-F'**) Average traces (**F**) and dot plots show that the average magnitude of apical (**F'**) ΔF/F GCaMP6s signals in mechanosensory hair bundles is not different in *cav1.3a$^{-/-}$* and *otofb$^{-/-}$* mutants compared to siblings. (**G-G'**) The spatial patterns of the evoked calcium influx (GCaMP6s ΔF, indicated via the heatmaps) at wildtype presynapses (**G'**) compared to prestimulus (**G**). (**H-H'**) Average traces (**H**) and dot plots show that the average magnitude of presynaptic (**H'**) ΔF/F GCaMP6s signals is absent in *cav1.3a$^{-/-}$* but unaltered in *otofb$^{-/-}$* mutants compared to siblings. (**I-I'**) The spatial patterns of evoked exocytosis (SypHy ΔF, indicated via the heatmaps) at sibling presynapses (**I'**) compared to prestimulus (**I**). (**J-J'**) Averaged traces (**J**) and dot plots show that presynaptic (**J'**) ΔF/F SypHy signals are absent in *cav1.3a$^{-/-}$* and in *otofb$^{-/-}$* mutants. The fluid-jet stimulus depicted as a gray box in F, H, and J. Each point in the dot plots represents one neuromast. All measurements were performed in mature neuromasts at 5–6 dpf on 3 animals and 9 neuromasts per genotype. Error bars: SEM. A one-way AVOVA with a Dunnett's correction for multiple tests was used in F' and J', and a Kruskal-Wallis test with a Dunn's correction for multiple tests was used in H'. ** $p<0.01$, **** $p<0.0001$. Scale bar = 5 μm.

The online version of this article includes the following source data and figure supplement(s) for figure 1:

**Source data 1.** Mean numbers and statistics for functional analyses.

**Figure supplement 1.** Spontaneous postsynaptic afferent activity is absent in *cav1.3a* mutant and *otofb* mutants.

**Figure supplement 1—source data 1.** Mean numbers and statistics for afferent spikes.

**Figure supplement 2.** Further characterization of memGCaMP6s responses in *cav1.3a* and *otofb* mutants.

**Figure supplement 2—source data 1.** Mean numbers and statistics for functional analyses.

## Results

### *Cav1.3a* and *otofb* mutations impair overlapping aspects of hair-cell neurotransmission

In sensory hair cells, the opening of presynaptic Ca$_v$1.3 channels enables calcium influx that initiates neurotransmission (*Figure 1D–D'*). This process is aided by the calcium sensor Otoferlin which couples calcium influx with vesicle exocytosis. Our work uses zebrafish models lacking functional Ca$_v$1.3 channels or Otoferlin to understand how neurotransmission impacts the hair cell's ability to respond to cellular stressors. Prior to examining cellular stressors in this system, we first thoroughly characterized how the lack of Ca$_v$1.3 channels or Otoferlin impacts neurotransmission in lateral-line hair cells.

Previous studies have shown that *ca$_v$1.3a* zebrafish mutants exhibit impaired hair-cell function as well as defects in hearing and balance (*Sidi et al., 2004*). Hearing and balance defects were also observed in zebrafish following morpholino knockdown of Otoferlin mRNA in hair cells (*Chatterjee et al., 2015*). Analysis of zebrafish *otof* morphants revealed that both *otofa* and *otofb* are present in hair cells in the zebrafish inner ear, while only *otofb* is present in hair cells of the lateral-line system. Based on this information, we generated stable *otofa* and *otofb* mutants using the CRISPR-Cas9 system (*Varshney et al., 2016*). In line with previous work, we found that Otoferlin immunolabel is absent in the lateral-line hair cells of *otofb* mutants (*Figure 1—figure supplement 1A-D*; *Varshney et al., 2016*). Moving forward we examined hair-cell activity in *ca$_v$1.3a* and *otofb* mutants using previously established methods and transgenic zebrafish lines expressing genetically encoded indicators (GEIs) of activity in hair cells (*Lukasz and Kindt, 2018*).

In hair cells, sensory stimuli deflect mechanosensory hair bundles leading to an apical influx of cations. This cation influx depolarizes the cell and triggers the influx of calcium through presynaptic Ca$_v$1.3 channels, followed by exocytosis of vesicles at the presynapse (*Figure 1D*). To measure mechanosensation-dependent and presynaptic calcium signals we used a membrane localized GCaMP (memGCaMP6s, hereafter referred to as GCaMP6s; *Figure 1B–C and E–H'*, *Figure 1— figure supplement 2A-L*), along with a fluid-jet to deflect hair bundles (*Lukasz and Kindt, 2018*). Previous work has shown that microphonic potentials (summed potentials dependent on mechanotransduction) are qualitatively reduced in Ca$_v$1.3a-deficient neuromasts (*Nicolson et al., 1998*; *Trapani and Nicolson, 2011*). Therefore, we first examined mechanosensation-dependent calcium signals in hair bundles. In response to a 500 ms fluid-jet stimulus, we observed that the magnitude, slope, and duration of GCaMP6s signals in apical mechanosensory bundles were comparable to

those observed in sibling controls in both *ca$_V$1.3a* and *otofb* mutants (**Figure 1E–F'**, **Figure 1— figure supplement 1B,C**). Because of the reported difference in microphonic potentials in *ca$_V$1.3a* mutants, we also examined the baseline GCaMP6s intensity in the hair bundles of *ca$_V$1.3a* and *otofb* mutants. We found that although it was not significantly different, the resting GCaMP6s intensity in hair bundles of *ca$_V$1.3a* mutants tended to be higher relative to controls (**Figure 1—figure supplement 2A**); no difference was observed in *otofb* mutants. In addition, we also observed that GCaMP6s responses returned to baseline faster in *ca$_V$1.3a* mutants relative to controls (**Figure 1F**, **Figure 1— figure supplement 2D**). No difference in return to baseline was observed in *otofb* mutants. When we used a shorter, 200 ms stimulus, we observed similar trends with regard to: magnitude, duration, slope, and recovery (**Figure 1—figure supplement 2E-H**). Because the resting calcium levels in hair bundles are higher in *ca$_V$1.3a* mutants and the recovery of the GCaMP6s responses is faster, it is possible that *ca$_V$1.3a* mutants have subtle defects in mechanotransduction. In the future, electrophysiological approaches are needed to assess mechanotransduction more carefully in both *ca$_V$1.3a* and *otofb* mutants. Overall, our GCaMP6s measurements indicate that both Ca$_V$1.3a- and Otofb-deficient hair cells exhibit largely normal evoked mechanotransduction. This result is important in light of previous work that found eliminating hair-cell mechanosensation could render hair cells more resistant to cellular stressors (**Pickett et al., 2018**). Thus, we are able to focus our studies on events downstream of hair-cell mechanosensation.

We next examined evoked GCaMP6s responses at the base of hair cells to characterize presynaptic calcium signals in *ca$_V$1.3a* and *otofb* mutants. Our measurements at the base of hair cells revealed that during a 500 ms stimulus, the magnitude, slope, duration, and recovery of GCaMP6s signals in *otofb* mutants was unchanged compared to controls (**Figure 1G–H'**, **Figure 1—figure supplement 2J-L**). In contrast, *cav1.3a* mutants completely lacked presynaptic-calcium responses (**Figure 1G–H'**). We also examined the resting GCaMP6s intensity at the base of hair cells and found that, consistent with a loss of Ca$_V$1.3-channel activity, resting calcium levels were significantly reduced in *ca$_V$1.3a* mutants compared to controls (**Figure 1—figure supplement 2I**); no difference was observed in *otofb* mutants. Our presynaptic measurements indicate that, similar to results obtained in mice, Ca$_V$1.3 channels but not Otoferlin are required for presynaptic-calcium influx in lateral-line hair cells.

We next measured evoked exocytosis at the hair-cell presynapse in *ca$_V$1.3a* and *otofb* mutants using the genetically encoded indicator (GEI) SypHy (**Figure 1I–J'**). SypHy is a pH sensitive GFP (pHluorin), targeted to the interior of synaptic vesicles via fusion to synaptophysin. The low pH inside synaptic vesicles quenches SypHy fluorescence, but upon exocytosis, the pH rises along with SypHy fluorescence (**Granseth et al., 2006**). Upon hair-cell stimulation, we observed no increase in SypHy fluorescence in either mutant, revealing that both mutants lack evoked exocytosis (**Figure 1I–J'**). This lack of exocytosis in zebrafish hair cells agrees with previously published data collected from the auditory system in Ca$_V$1.3 and Otof mouse mutants (**Brandt et al., 2003**; **Roux et al., 2006**). While Otoferlin is essential for exocytosis in auditory hair cells, residual evoked and spontaneous exocytosis has been observed in the vestibular system of *Otof* mouse mutants (**Dulon et al., 2009**). Because of this discrepancy we used loose patch-clamp electrophysiology to quantify spontaneous afferent activity in *ca$_V$1.3a* and *otofb* zebrafish mutants. In the lateral line, spontaneous spikes detected in afferent neurons (in the pLLg; **Figure 1A**) result from the spontaneous fusion of vesicles at the hair-cell synapse (**Trapani and Nicolson, 2011**). In agreement with previously published data, we detected very few spontaneous afferent spikes in *ca$_V$1.3a* mutants (**Trapani and Nicolson, 2011**; **Figure 1—figure supplement 1E-F**). We also examined *otofb* mutants and similarly observed very few spontaneous afferent spikes (**Figure 1—figure supplement 1E-F**; sibling: 399.0±79.7, *ca$_V$1.3a*: 5.9±2.9, *otofb*: 4.2±1.2 spikes per min). Overall, our SypHy and electrophysiology results indicate that in lateral-line hair cells, both Ca$_V$1.3 channels and Otoferlin are essential for evoked and spontaneous exocytosis.

Together our comprehensive functional assessment of *ca$_V$1.3a* and *otofb* mutants revealed that both mutants have relatively normal mechanosensitive function. In lateral-line hair cells Ca$_V$1.3 channels are required for both presynaptic calcium influx and exocytosis while Otoferlin is only required for exocytosis. Importantly, *ca$_V$1.3a* and *otofb* mutants impair overlapping aspects of hair-cell neurotransmission and are useful models to explore the relationship between neurotransmission and metabolic stress.

# Chronic loss of neurotransmission protects hair cells against the ototoxin neomycin

In neurons, the energy demands of neurotransmission can lead to a buildup of cytotoxic metabolic byproducts and render cells more susceptible to insults (*Singh et al., 2019*). Our functional results demonstrate that loss of Ca$_V$1.3a or Otofb function impairs overlapping aspects of neurotransmission in hair cells. How these particular aspects of neurotransmission impact hair-cell health and response to stressors is not known. Hair cells are acutely susceptible to many stressors, including ototoxins such as aminoglycoside antibiotics (*Bitner-Glindzicz and Rahman, 2007*; *Coffin et al., 2010*). We used *ca$_V$1.3a* or *otofb* mutants to explore how neurotransmission impacts the susceptibility of hair cells to a known stressor, the aminoglycoside antibiotic neomycin.

For our study, we challenged Ca$_V$1.3a and Otofb-deficient hair cells in larvae with functional lateral-line systems (at 5 or 6 days post fertilization (dpf)) with varying concentrations of neomycin: 75, 100, and 200 μM. Prior to neomycin treatment, we immobilized larvae and used a fluorescent reporter line (GCaMP6s) to quantify the total number of hair cells in each neuromast (*Figure 2A*). We then incubated larvae with neomycin for 30 min. After this incubation, we washed off the neomycin and used the styryl dye FM 4–64 to label and help identify surviving cells (*Figure 2A'–C*). Using both our fluorescent reporter and FM 4–64 label to reliably identify surviving hair cells per neuromast, we found that overall, in both *cav1.3a* and *otofb* mutants, hair-cell survival was augmented relative to sibling controls (*Figure 2A–E*). At all concentrations tested, the mean % hair-cell survival for *cav1.3a* and *otofb* mutants was significantly higher compared to sibling controls. These results indicate that the neurotransmission defects in *cav1.3a* and *otofb* mutants impart resistance to neomycin.

The resistance to neomycin observed in *cav1.3a* and *otofb* mutants occurred after a chronic loss of neurotransmission. Therefore, we investigated whether transient block of neurotransmission could also augment hair-cell survival. For this work we used a pharmacological approach. We first blocked Ca$_V$1.3 channels with the L-type calcium channel antagonist isradipine (*Fitton and Benfield, 1990*). Previous work has demonstrated that isradipine blocks presynaptic calcium influx without impairing mechanotransduction (*Zhang et al., 2018*). For this experiment we preincubated larvae in 10 μM isradipine for 10 min, followed by a 30 min co-incubation with neomycin. After acute isradipine treatment, we observed no change in survival with a mean % hair-cell survival comparable to DMSO-treated controls (*Figure 2F*, 5-6 dpf). After we observed no protection using an acute isradipine application, we decided to switch to longer incubations. Here, we applied 10 μM isradipine to larvae at 4 dpf for 24 hr or at 3 dpf for 48 hrs prior to 100 μM neomycin treatment at 5 dpf. In contrast to our acute isradipine application, both of the longer 24 and 48 hr isradipine incubations significantly augmented hair-cell survival after neomycin treatment (*Figure 2G-H*, 5 dpf). This indicates that a relatively long (~24 hr) block of presynaptic calcium influx can confer significant resistance to neomycin.

Similar to the genetic disruption in *cav1.3a* mutants, isradipine treatment blocks not only presynaptic calcium influx but also downstream exocytosis. Therefore, we used pharmacology to acutely and specifically disrupt the synaptic vesicle cycle. To disrupt the synaptic vesicle cycle, we used the dynamin inhibitor Dynole 34–2, which blocks endocytosis (*Jackson et al., 2015*). Previous work in neurons has shown that compensatory endocytosis is required to maintain a readily releasable pool of synaptic vesicles (*Wu et al., 2014*). To confirm Dynole 34–2 impairs hair-cell exocytosis without altering other aspects of hair-cell function, we performed several controls. Using calcium imaging, we found that 2.5 μM Dynole 34–2 treatment did not impair evoked mechanotransduction or presynaptic calcium influx compared to DMSO-treated controls (*Figure 2—figure supplement 1A-D*). Next, we recorded spontaneous spikes from hair-cell afferents to determine if Dynole 34–2 impairs exocytosis. Importantly, we found that after a 10-min treatment with 2.5 μM Dynole 34–2, the number of spontaneous spikes recorded from lateral-line afferents was significantly reduced (*Figure 2—figure supplement 1E-F*). Overall, this functional assessment indicates that Dynole 34–2 specifically impairs hair-cell exocytosis without any detectable impact on hair-cell mechanotransduction or presynaptic-calcium influx. After verifying the efficacy of Dynole 34–2, we applied 2.5 μM Dynole 34–2 for the duration of the 30 min neomycin treatment. Like our isradipine experiments, for this transient Dynole 34–2 treatment, we observed no significant protection with a mean % hair-cell survival similar to DMSO-treated controls (*Figure 2F*, 5-6 dpf). However, pretreatment with 2.5 μM Dynole 34–2 for 24 or 48 hr prior to neomycin treatment offered significant protection compared to DMSO controls (*Figure 2G-H*, 5 dpf).

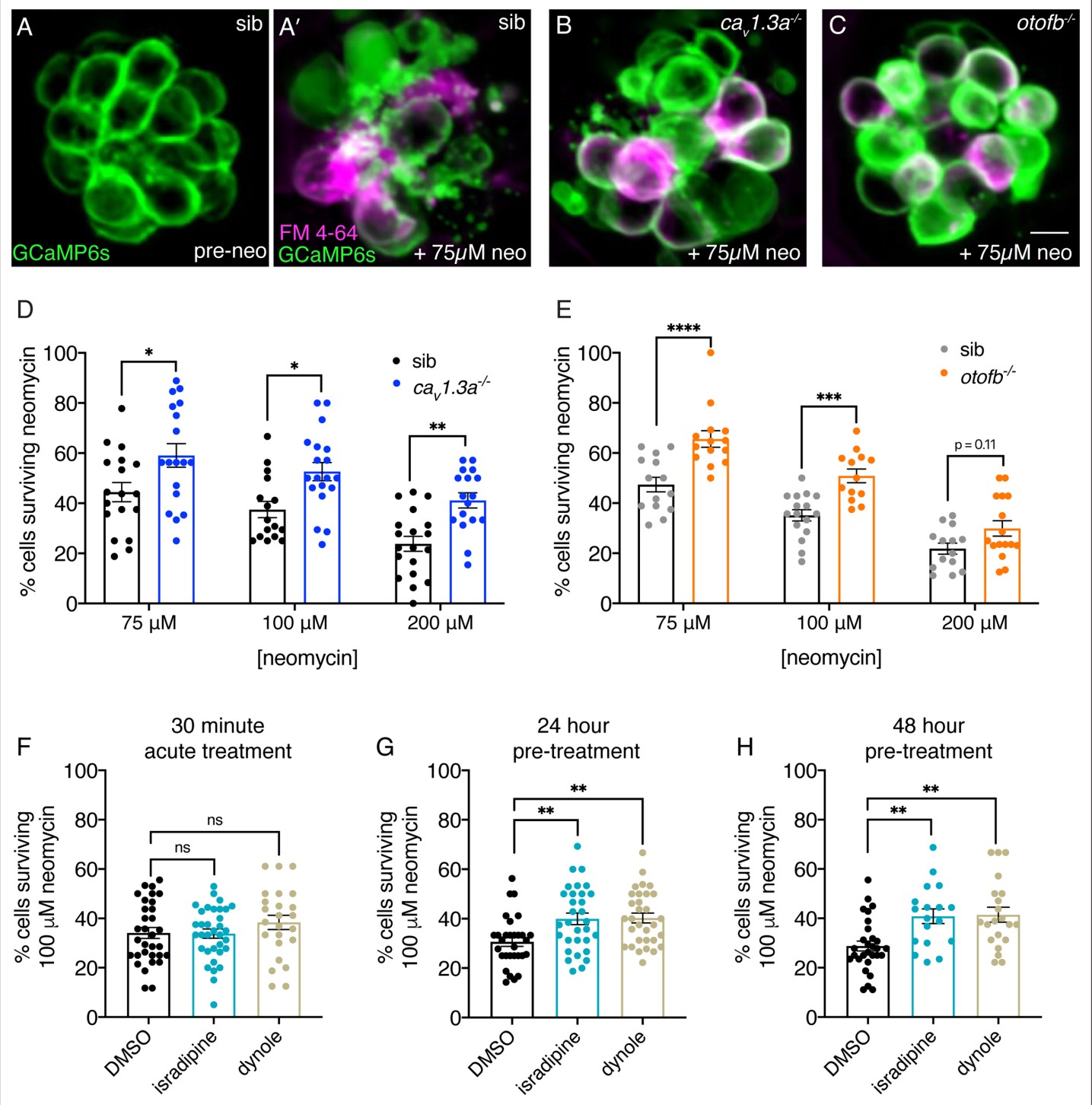

**Figure 2.** Chronic loss of neurotransmission protects hair cells from the ototoxin neomycin. (**A-A′**) Hair cells in siblings before (**A**) and after a 30-min treatment with 75 µM neomycin (**A′**). GCaMP6s outlines hair cells. The presence of FM 4–64 in hair cells reveals surviving cells (**A′**). (**B-C′**) After a 30-min treatment with 75 µM neomycin GCaMP6s and FM 4–64 more surviving hair cells are found in *cav1.3a⁻ᐟ⁻* (**B**), and *otofb⁻ᐟ⁻* (**C**) mutants. (**D–E**) A higher percentage of Caᵥ1.3- and Otof-deficient hair cells survive a 30 min treatment with three different neomycin concentrations (75, 100, and 200 µM) compared to sibling controls. For quantification in D-E, neuromasts were examined at 5 or 6 dpf immediately after washout of neomycin solution and application of FM 4–64. (**F**) Percentage of hair cells per neuromast surviving is not altered when hair cells in wildtype larvae are co-incubated with 0.1% DMSO (control), 10 µM isradipine, or 2.5 µM Dynole 34–2 during the 30-min 100 µM neomycin treatment. (**G–H**) When wildtype hair cells are incubated with 10 µM isradipine or 2.5 µM Dynole 34–2 for 24 hr (G, 4 to 5 dpf) or for 48 hr (H, 3 to 5 dpf) prior to neomycin treatment significantly more hair cells survive compared to DMSO controls. Each point in the dot plots in D-H represents one neuromast. A minimum of five animals were examined per

*Figure 2 continued on next page*

*Figure 2 continued*

treatment group. Error bars: SEM. For comparisons, a two-way ANOVA with a Sidak's correction for multiple comparisons was used in D-E. A one-way AVOVA with a Dunnett's correction for multiple comparisons was used in F, G and H. * p<0.05, ** p<0.01, *** p<0.001. Scale bar = 5 μm.

The online version of this article includes the following source data and figure supplement(s) for figure 2:

**Source data 1.** Mean numbers and statistics for ototoxicity analyses.

**Figure supplement 1.** Treatment with 2.5 μM Dynole 34–2 reduces the frequency of afferent spiking and does not affect mechanotransduction or presynaptic calcium influx.

**Figure supplement 1—source data 1.** Mean numbers and statistics for functional analyses.

**Figure supplement 2.** *Cav1.3a* and *otofb* mutants are resistant to gentamicin-induced cell death.

**Figure supplement 2—source data 1.** Mean numbers and statistics for ototoxicity analyses.

Taken together, analysis of our mutant models and our pharmacology experiments suggest that prolonged loss (~24 hr) of neurotransmission reduces neomycin susceptibility. Surprisingly, impairing the synaptic vesicle cycle even in the presence of normal calcium currents results in a level of neomycin resistance comparable to that observed in the absence of both calcium influx and exocytosis. This suggests that it is the synaptic vesicle cycle contributes to hair-cell neomycin susceptibility.

## Chronic loss of Cav1.3a or Otofb protects hair cells against the ototoxin gentamicin

Neomycin is just one of many cytotoxic stressors that can be acutely damaging to hair cells. Although they belong to the same family of antibiotics, neomycin and gentamicin induce hair-cell death along distinct time courses (*Hailey et al., 2017*; *Owens et al., 2009*). In hair cells, gentamicin-induced cell death occurs over a period of hours, whereas neomycin-induced cell death can begin within minutes. Based on these divergent timescales it has been proposed that these two antibiotics may also work through different cellular pathways to induce cell death. To understand if neurotransmission contributes more generally to ototoxin susceptibility, we tested whether hair cells in *cav1.3a* and *otofb* mutants are also less susceptible to the aminoglycoside antibiotic gentamicin.

Previous studies have indicated that gentamicin-induced hair-cell loss may continue to occur even up to 24 hr post exposure (*Owens et al., 2009*). Due to this longer time course of gentamicin ototoxicity (and inability to keep the animals restrained for the entire duration), we used a different method to quantify hair-cell survival compared to our neomycin experiments. For our gentamicin experiments, we treated *cav1.3a* and *otofb* mutants and siblings with 200 μM gentamicin for 2 hr at 5 dpf. We then washed off the gentamicin, waited 24 hr, and assessed hair-cell survival at 6 dpf (*Figure 2—figure supplement 2A*). Before assessing cell survival at 6 dpf, we stained with FM 4–64 to identify surviving cells (*Figure 2—figure supplement 2B, D-F*). We then quantified the number of hair cells present in gentamicin-treated mutants and sibling controls. We also quantified the number of hair cells present in untreated mutants and sibling controls. We then normalized the number of cells present in treated animals to the number of cells present in untreated animals for each genotype to obtain the % of surviving hair cells. Using this approach, after gentamicin treatment, we found significantly more surviving cells in both *ca$_v$1.3a* and *otofb* mutants relative to controls (*Figure 2—figure supplement 2B-F*). Together, our neomycin and gentamicin assays indicate that neurotransmission may impact the susceptibility of hair cells to a diverse range of ototoxins.

## *Cav1.3a* and *otofb* mutant hair cells have normal neomycin uptake and clearance

To exert their toxicity, aminoglycoside antibiotics such as neomycin and gentamicin enter hair cells primarily via mechanosensitive ion channels in hair bundles (*Figure 1D*; *Alharazneh et al., 2011*). Our functional assessment of *ca$_v$1.3a* and *otofb* mutants did not reveal any major defects in hair-cell mechanosensation (*Figure 1E–F'*). Regardless of this assessment, we wanted to ensure that the impaired neurotransmission in *ca$_v$1.3a* or *otofb* mutant hair cells did not impact neomycin or gentamicin uptake sufficiently to confer protection.

Therefore, we examined neomycin uptake in Ca$_v$1.3a- or Otofb-deficient hair cells. To investigate neomycin uptake in hair cells, we conjugated neomycin to the fluorophore Texas Red (Neo-TR) as

described previously (*Stawicki et al., 2014*). We examined the kinematics of neomycin uptake by collecting Z-stacks of hair cells every 60 s during a 10-min incubation in 25 µM Neo-TR. We quantified Neo-TR fluorescence at each time point. We observed no significant difference in the Neo-TR intensity between Ca$_V$1.3a- or Otofb-deficient hair cells and their respective sibling controls at the 10-min endpoint (*Figure 3A–B*). We also observed no significant difference between mutants and siblings at any timepoint during Neo-TR uptake (*Figure 3C and E*). These results indicate that lack of Ca$_V$1.3 channels or Otoferlin does not impair the ability of neomycin to enter hair cells.

Although uptake of Neo-TR was unaffected in *ca$_V$1.3a* or *otofb* mutants, it is possible that defects in neurotransmission could offer protection via the augmented clearance of neomycin. Previous studies have found that neomycin preferentially accumulates within hair cells and is not rapidly cleared (*Hailey et al., 2017*; *Steyger et al., 2003*). We tested whether increased clearance of neomycin could account for a reduced susceptibility in our mutants. At the end of the 10-min Neo-TR uptake period, we washed out the Neo-TR. After washout, we collected Z-stacks every 90 s for 30 min to visualize Neo-TR clearance from hair cells. While we did observe a decrease in Neo-TR signal in both *ca$_V$1.3a* and *otofb* mutant hair cells, this decrease was not significantly different compared to sibling controls (*Figure 3D and F*). Similar Neo-TR clearance trajectories indicate that the protection in in *ca$_V$1.3a* or *otofb* mutants is not due to augmented neomycin clearance.

After uptake into hair cells, Neo-TR rapidly forms puncta. Previous work in the lateral line has shown that these puncta co-label with the lysosomal markers LysoTracker green and GFP-Rab7 (*Hailey et al., 2017*). Loading of neomycin into these lysosomes is proposed to be a protective measure. Therefore, it is possible that *ca$_V$1.3a* or *otofb* mutants may show differences in lysosomal loading. Close examination of Neo-TR label during uptake did not reveal any striking differences in lysosomal loading. The presence of bright puncta, indicative of Neo-TR sequestration within lysosomes, is apparent in both *ca$_V$1.3a* and *otofb* mutants (*Figure 3—figure supplement 1*). The lack of gross differences in the observed pattern of formation of Neo-TR lysosomal puncta suggests that lack of Ca$_V$1.3 channels or Otoferlin does not offer protection through differential lysosomal loading. Taken together, our assays using Neo-TR indicate that the neomycin protection observed in Ca$_V$1.3- or Otofb-deficient hair cells is not due to altered Neo-TR uptake, clearance, or trafficking into lysosomes.

## Hair cells that exhibit evoked neurotransmission are younger and more resistant to neomycin

Previous work by our group demonstrated while all hair cells in each lateral-line neuromast are mechanosensitive, only a subset of cells exhibit neurotransmission when depolarized by mechanical stimuli (*Figure 1E–E', G–G'1–I'*; *Zhang et al., 2018*). Our current study indicates that a loss of neurotransmission may offer protection against hair-cell stressors such as neomycin. Therefore, we sought to investigate whether the presence or absence of neurotransmission within neuromasts of wildtype hair cells correlates with neomycin susceptibility.

For these experiments, we first used functional calcium imaging to identify hair cells with (active) and without (inactive) evoked presynaptic calcium influx (*Figure 4A–A'*; stars indicate active cells exhibiting evoked neurotransmission). After identifying cells based on presynaptic function, we then applied 75 or 100 µM neomycin for 30 min. Surprisingly, after neomycin treatment, we observed that active hair cells exhibit significantly higher survival compared to inactive hair cells (example: *Figure 4A–A'''*, quantification: *Figure 4B*). This indicates that within wildtype neuromasts, the presence of neurotransmission is associated with protection from neomycin.

This observation presented an unexpected complexity in our current study. On one hand, we found that perturbations that result in chronic loss of neurotransmission in all hair cells confer protection from ototoxins. On the other hand, within wildtype hair cells, the presence of evoked neurotransmission correlates with protection from ototoxins compared to cells without evoked neurotransmission. Moving forward we explored what could explain this discrepancy. Numerous studies have shown that neuromasts rapidly acquire new cells over the course of development (2–4 dpf). By the time the lateral-line system becomes functional around 5 dpf, each neuromast contains cells of different ages. In addition, previous work has suggested that hair cells of different ages show differences in susceptibility to neomycin–namely that older hair cells are thought to exhibit higher susceptibility (*Murakami et al., 2003*; *Pickett et al., 2018*). Based on this evidence we directly tested whether surviving hair cells with evoked neurotransmission at 5 dpf were in fact younger.

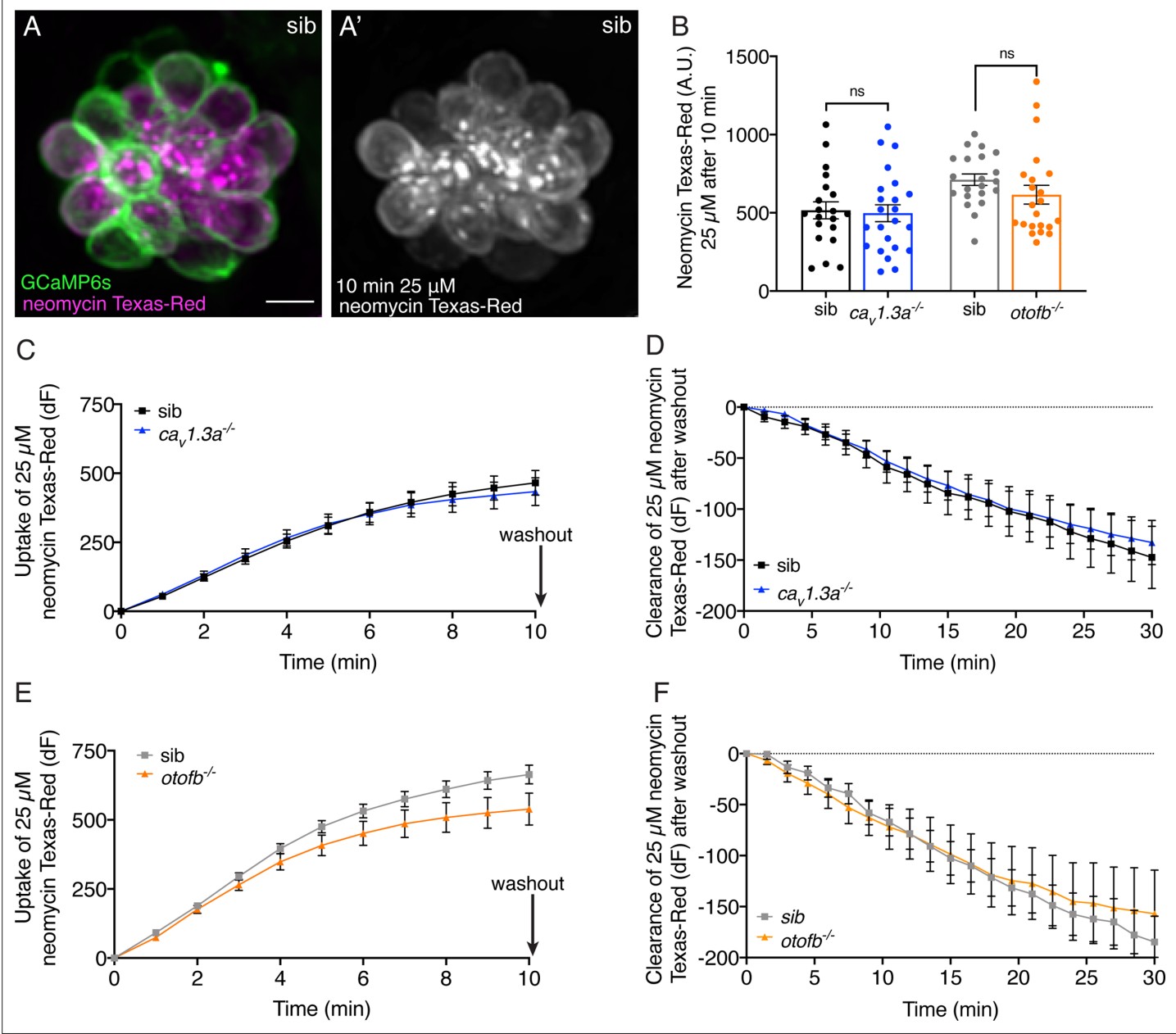

**Figure 3.** *Cav1.3a* and *otofb* mutant hair cells have normal neomycin uptake and retention. (**A-A'**) Example of hair cells in a wildtype neuromast after incubation in 25 μM neomycin-Texas Red solution (Neo-TR) for 10 min. GCaMP6s outlines hair cells in A. (**B**) Average dot plots of Neo-TR fluorescence intensity in neuromasts from *cav1.3a*−/− (blue) and *otofb*−/− (orange) mutants and siblings (black and grey, respectively) after a 10-min incubation in 25 μM Neo-TR. No differences in overall neomycin uptake are seen between mutants and siblings. (**C–E**) Average Neo-TR fluorescence intensity over the course of 10 min 25 μM exposure in *cav1.3a*−/− mutant neuromasts (blue) and siblings (black) (**C**) and *otofb*−/− mutant neuromasts (orange) and siblings (grey) (**E**). No differences are seen between mutants and siblings in the time course of neomycin uptake. (**D–F**) Average Neo-TR fluorescence intensity over the course of 30 min following Neo-TR washout in *cav1.3a*−/− mutant neuromasts (blue) and siblings (black) (**D**) and *otofb*−/− mutant neuromasts (orange) and siblings (grey) (**F**). No significant differences were detected in neomycin clearance or retention between mutants and siblings. Each point in B represents one neuromast. A minimum of 14 neuromasts are averaged for the plots in C-F. A minimum of five animals at 5 dpf were examined per treatment group. Error bars: SEM. For comparisons, unpaired t-tests were used in B, and a two-way ANOVA with a Sidak's correction for multiple comparisons was used in C-F. Scale bar = 5 μm.

The online version of this article includes the following source data and figure supplement(s) for figure 3:

**Source data 1.** Mean numbers and statistics for ototoxicity analyses.

**Figure supplement 1.** Time course of neomycin uptake and packaging in *cav1.3a* and *otofb* mutants and siblings.

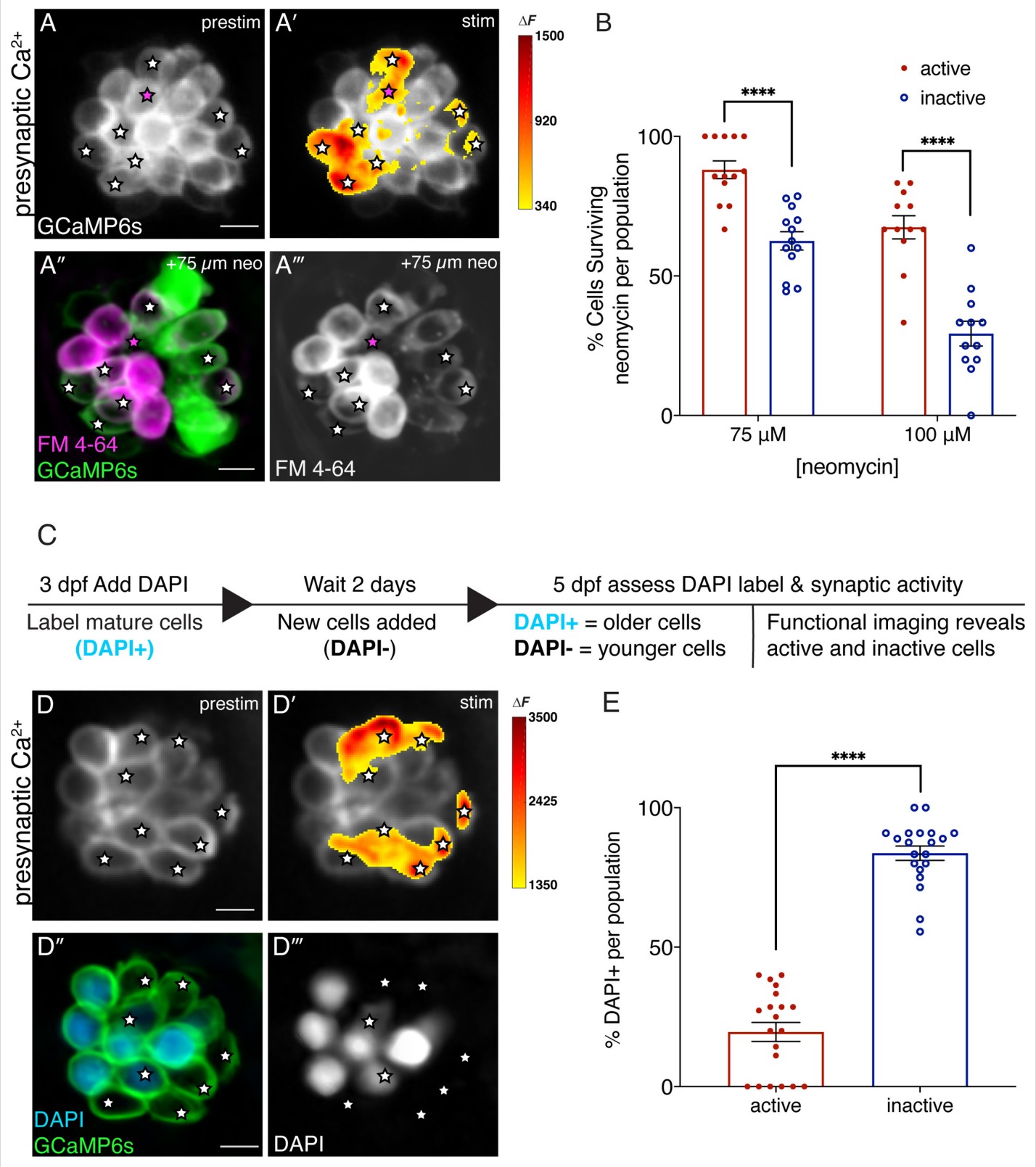

**Figure 4.** Synaptically active cells are relatively young and more resistant to neomycin. (**A–A′**) Hair cells from wildtype 5 dpf fish before and during a 2 s fluid-jet stimulus. The spatial patterns of the evoked calcium influx during stimulation (A′ GCaMP6s ΔF, indicated via the heatmaps) compared to prestimulus (**A**) reveal synaptically active cells (white stars). (**A″–A‴**) Hair cells depicted in A and A′ after a 30-min treatment with 75 µM neomycin solution. Magenta star demarcates an active cell that did not survive neomycin treatment. (**B**) A higher percentage of active hair cells survive a 30-min

*Figure 4 continued on next page*

*Figure 4 continued*

treatment with two different neomycin concentrations (75 and 100 µM) compared to inactive cells. Neuromasts were examined at 5 dpf immediately after washout of neomycin solution and application of FM 4–64. (**C**) Outline of DAPI labeling protocol used to differentiate between older and younger cells within a given neuromast. DAPI is used to label mature hair cells at 3 dpf. At 5 dpf, the hair cells are assessed. (**D-D'**) Hair cells from wildtype 5 dpf fish before and during a 2 second fluid-jet stimulus. The spatial patterns of the evoked calcium influx during stimulation (D' GCaMP6s ΔF, indicated via the heatmaps) compared to prestimulus (**D**) reveal synaptically active cells (white stars). (**D"-D'''**) Hair cells depicted in D and D' showing DAPI-positive older and DAPI-negative younger cells. (**E**) DAPI-positive older cells make up a much greater percentage of the inactive cell population than the active cell population. A minimum of five animals were examined per experimental group at 5 dpf. Each dot in B and E represents one neuromast. Error bars: SEM. For comparisons, a two-way ANOVA with a Sidak's correction for multiple comparisons was used in B. A Mann-Whitney test was used in E. **** p<0.0001. Scale bar = 5 µm.

The online version of this article includes the following source data and figure supplement(s) for figure 4:

**Source data 1.** Mean numbers and statistics for activity-state analyses.

**Figure supplement 1.** Older hair cells are more susceptible to neomycin.

**Figure supplement 1—source data 1.** Mean numbers and statistics for cell stage and survival.

To differentiate hair cells based on age, we used two approaches: DAPI labeling and kinocilial height measurements. For DAPI labeling, we applied the vital dye at 3 dpf. At this stage, DAPI is only taken up by functionally mature cells via mechanosensitive ion channels. Importantly, similar to previous work using Hoechst label, we observed that DAPI is retained in the same set of hair cells days after dye labeling (*Pickett et al., 2018*). Thus, the presence of DAPI label at 5 dpf can distinguish older cells (DAPI-positive) from younger cells (DAPI-negative) that matured or were added to the neuromast after the initial DAPI labeling at 3 dpf (*Figure 4—figure supplement 1A-B'*). In addition to DAPI labeling, we measured the height of the tallest part of the hair bundle, the kinocilium, at 5 dpf. Previous work has shown that kinocilial height is a useful way to stage hair cells developmentally (*Kindt et al., 2012*). After DAPI labeling at 3 dpf, we found that the kinocilial heights of all DAPI-positive hair cells at 5 dpf were >20 µm, consistent with a mature population of hair cells (*Figure 4—figure supplement 1C*). In contrast DAPI-negative hair cells had a more even spread of kinocilial heights (2.5–23.5 µm), consistent with a developing population of hair cells (*Figure 4—figure supplement 1C*). Overall, these data indicate that either kinocilial height or DAPI labeling can be used to estimate hair-cell age. Moving forward we used kinocilial height measurements to determine whether older hair cells were more susceptible to neomycin (*Figure 4—figure supplement 1D*). After application of 100 µM neomycin for 30 min, we observed that younger hair cells had significantly higher survival compared to mature hair cells (*Figure 4—figure supplement 1E-E'*; % of young cells surviving: 63.17%±8.62, % of mature cells surviving: 10.93%±3.16). This indicates that within wildtype neuromasts, younger hair cells are indeed less susceptible to neomycin.

Next, we investigated whether cells that exhibit evoked neurotransmission are more resistant to neomycin because they are relatively younger. For this work we labeled hair cells with DAPI at 3 dpf and then used functional calcium imaging at 5 dpf to identify hair cells with (active) and without (inactive) evoked presynaptic calcium influx. We found that significantly more inactive cells were older and DAPI-positive compared to active cells (*Figure 4D–D'''*, *Figure 4E*; active: 22.5%, inactive 87.5% cells per neuromast DAPI-positive). Overall, our DAPI experiments revealed that within a given neuromast, active hair cells exhibiting evoked neurotransmission represent a younger population of hair cells compared to inactive hair cells. Furthermore, their relative youth helps to explain how hair cells with evoked neurotransmission are more resistant to neomycin despite our evidence that chronic loss of neurotransmission is protective against neomycin.

## $Ca_v1.3a$ and *otofb* mutants exhibit lower baseline mitochondrial oxidation

The lack of neurotransmission we observed in older hair cells was intriguing (*Figure 4C–E*). It is also consistent with the hypothesis that over time, activity-driven ATP synthesis in hair cells can lead to the accumulation of reactive oxygen species (ROS) and induce cellular damage. If neurotransmission is a driver of these metabolic demands, synaptic silencing in older cells could be a way to prevent ROS accumulation and associated pathology. But whether neurotransmission correlates with an increase in ROS in hair cells is not known.

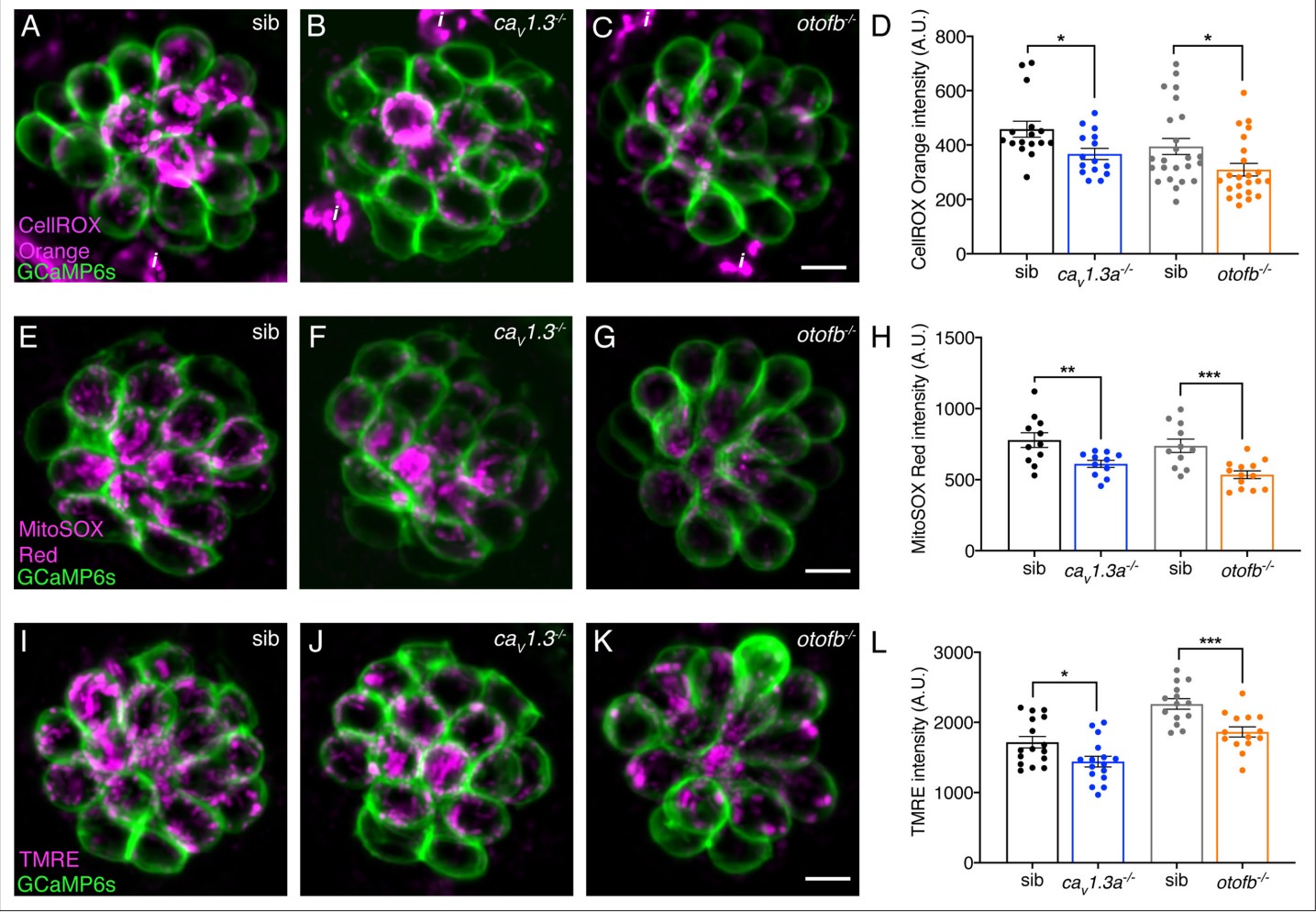

**Figure 5.** *Ca_v 1.3a* and *otofb* mutants exhibit reduced mitochondrial oxidation and mitochondrial activity. (**A–C**) Hair cells in a wildtype, *cav1.3a^{-/-}* or *otofb^{-/-}* neuromast (labeled with GCaMP6s) after 30-min incubation with 12.5 µM CellROX. (**B**) Average dots plots show that CellROX Orange fluorescence intensity is lower in *cav1.3a^{-/-}* (blue) and *otofb^{-/-}* (orange) mutants compared to wildtype siblings (black, gray). (**E–G**) Hair cells in a wildtype, *cav1.3a^{-/-}* or *otofb^{-/-}* neuromast (labeled with GCaMP6s) after a 15-min incubation with 5 µM MitoSOX. (**H**) Average dots plots show that MitoSOX Red fluorescence intensity is lower in *cav1.3a^{-/-}* (blue) and *otofb^{-/-}* (orange) mutants compared to wildtype siblings (black, gray). (**I–K**) Hair cells in a wildtype, *cav1.3a^{-/-}* or *otofb^{-/-}* neuromast (labeled with GCaMP6s) following a 30-min incubation with 10 nM TMRE. (**L**) Average dots plots show that the TMRE fluorescence intensity is reduced in both *cav1.3a^{-/-}* (blue) and *otofb^{-/-}* (orange) mutants compared to respective siblings (black, gray). Each dot in D, H, and L represents one neuromast. A minimum of 5 animals were examined at 6 dpf per treatment group. Error bars: SEM. For comparisons, an unpaired t-test was used. *i* in A-C are not hair cells but ionocytes labeled by CellROX. * $p<0.05$, ** $p<0.01$, *** $p<0.001$. Scale bar = 5 µm.

The online version of this article includes the following source data and figure supplement(s) for figure 5:

**Source data 1.** Mean numbers and statistics for mitochondrial dye labeling.

**Figure supplement 1.** Hair cell development and turnover are largely normal in *ca_v1.3a* and *otofb* mutants.

**Figure supplement 1—source data 1.** Mean numbers and statistics for cell stage and turnover.

**Figure supplement 2.** Evoked mitochondrial calcium uptake is normal in *otofb* mutants and absent in *ca_v1.3a* mutants.

**Figure supplement 2—source data 1.** Mean numbers and statistics for mitoGCaMP3 responses.

**Figure supplement 3.** JC-1 shows a trend towards reduced mitochondrial activity in *ca_v1.3a* and *otofb* mutants.

**Figure supplement 3—source data 1.** Mean numbers and statistics for JC-1 labeling.

To investigate the relationship between ROS accumulation and neurotransmission, we applied the ROS indicator dyes CellROX Orange and MitoSOX Red to *ca_v1.3a* or *otofb* mutant hair cells. CellROX Orange is a general indicator of cellular oxidation and ROS levels, while MitoSOX Red is an indicator used to specifically detect mitochondrial superoxide. We found significantly lower CellROX Orange

baseline fluorescence in both $ca_V1.3a$ and *otofb* mutant hair cells relative to controls (*Figure 5A–D*). Likewise, labeling using MitoSOX Red revealed significantly lower MitoSOX Red baseline fluorescence in $ca_V1.3a$ and *otofb* mutant hair cells relative to controls (*Figure 5E–H*). Together our CellROX Orange and MitoSOX Red measurements suggest that the loss of neurotransmission in $Ca_V1.3a$ or Otofb-deficient hair cells results in a lower baseline level of ROS accumulation and lower oxidative stress. ROS are known to impart damage to DNA, lipids, proteins, and many other molecules throughout the cell, rendering cells overall more susceptible to insults (*Yang and Lian, 2020*). Therefore, lower levels of baseline ROS could explain why $ca_V1.3a$ and *otofb* mutant hair cells exhibit resistance to neomycin-induced cell death.

### $Ca_V1.3a$ and *otofb* mutants show relatively normal maturation and no hair-cell loss

Our ROS measurements indicate that $ca_V1.3a$ and *otofb* mutants are less susceptible to neomycin due to less accumulated ROS. Work in mice has shown that $Ca_V1.3a$-deficient auditory hair cells in mice retain immature characteristics (*Brandt et al., 2003*; *Eckrich et al., 2019*). Our results indicate that in the lateral line, younger hair cells are less susceptible to neomycin (*Figure 4—figure supplement 1E'*). In addition, mutations that impact hair-cell function in mice can not only impair development, but also lead to cell death (*Schwander et al., 2009*). In the lateral line, hair cells regenerate after cell death, and the death of many hair cells may result in higher rates of cell turnover (*Harris et al., 2003*). Therefore, it is possible that in the lateral line, $Ca_V1.3a$ or Otofb-deficient hair cells may appear to accumulate less ROS because they are younger–either due to incomplete maturation or higher rates of turnover.

To examine whether $Ca_V1.3a$ or Otofb-deficient hair cells have more turnover we first counted the number of hair cells per neuromast in developing or mature neuromasts at 3 and 5 dpf, respectively. This analysis revealed the same number of hair cells per neuromast in $ca_V1.3a$ and *otofb* mutants compared to controls at both ages (*Figure 5—figure supplement 1B,E*). Similar cell counts suggest that both mutants generate hair cells at a normal rate and show no obvious loss of hair cells. To further examine whether the hair cells in $ca_V1.3a$ or *otofb* mutants undergo more cell death or turnover, we used DAPI labeling to follow the same set of hair cells over time (*Figure 5—figure supplement 1A*). We first applied DAPI at 3 dpf to label and assess the number of functionally mature hair cells in each neuromast. Then we waited 2 days and reassessed the number of DAPI-positive cells remaining in the same neuromasts. We found that from 3 to 5 dpf, similar to controls, $Ca_V1.3a$ and Otofb-deficient hair cells did not show a significant loss of DAPI-positive hair cells (*Figure 5—figure supplement 1D,G*). Our DAPI results indicate that in the lateral line, loss of $Ca_V1.3a$ or Otofb does not result in more cell death or subsequently more cell turnover.

Lastly, we examined the developmental progression of hair cells in $ca_V1.3a$ and *otofb* mutants at 5 dpf using the height of the tallest part of the hair bundle, the kinocilium, to assess maturation (*Figure 5—figure supplement 1A*; *Kindt et al., 2012*). We found that the average kinocilial height was not different between $ca_V1.3a$ or *otofb* mutants compared to controls (*Figure 5—figure supplement 1C,F*). In addition, using the kinocilial height measurements, we split the cells into young and mature hair-cell populations (young <20 μm; mature ≥20 μm). After this split, we found that the average height of the kinocilium in young and mature hair cell populations was not different in $ca_V1.3a$ or *otofb* mutants compared to controls (*Figure 5—figure supplement 1C, F*). Our measurements of hair-bundle height suggest that hair cells in $ca_V1.3a$ and *otofb* mutants develop at a normal rate. Overall, our assessment revealed that $Ca_V1.3a$ or Otofb-deficient hair cells mature at a relatively normal rate and show no evidence of increased cell death or turnover. These results support the conclusion that $Ca_V1.3a$ or Otofb-deficient hair cells accumulate less ROS due to the loss of neurotransmission rather than due to cellular immaturity.

### Reduced mitochondrial potential but not reduced mitochondrial-calcium uptake is associated with neomycin resistance in $ca_V1.3a$ and *otofb* mutants

In both neurons and hair cells, presynaptic calcium influx can drive calcium into mitochondria (*Marland et al., 2016*; *Wong et al., 2019*). Furthermore, in neurons, calcium influx into mitochondria has been shown to stimulate oxidative phosphorylation in mitochondria and contribute to metabolic stress

(*Brookes et al., 2004*; *Tarasov et al., 2012*). Previous work has shown that when the lateral-line system matures (5 dpf), hair-cell stimulation drives calcium into mitochondria (*Pickett et al., 2018*; *Wong et al., 2019*). Therefore, we sought to investigate whether alterations in evoked mitochondrial calcium influx could explain why $ca_V1.3a$ or *otofb* mutants are less susceptible to neomycin-induced cell death.

We measured evoked mitochondrial-calcium uptake using a mitochondria-localized GCaMP3 (mitoGCaMP3) (*Figure 5—figure supplement 2A*). During fluid-jet stimulation, in mature hair cells at 5–6 dpf, we observed an absence of evoked mitochondrial-calcium uptake in $Ca_V1.3a$-deficient hair cells (*Figure 5—figure supplement 2B-B'*). This is consistent with previous pharmacological results showing that block of $Ca_V1.3$ channels with the antagonist isradipine inhibits mitochondrial-calcium uptake (*Wong et al., 2019*). Interestingly, during fluid-jet stimulation, we observed robust mitochondrial-calcium uptake in Otofb-deficient hair cells (*Figure 5—figure supplement 2C-C'*). We quantified the peak evoked mitoGCaMP3 responses in *otofb* mutants and siblings and found no significant difference (*Figure 5—figure supplement 2C'*). Overall, our mitoGCaMP3 measurements indicate that a reduction in evoked mitochondrial-calcium uptake is not a common factor leading to a reduction in neomycin susceptibility in Otofb- and $Ca_V1.3a$-deficient hair cells.

Although calcium influx into mitochondria can stimulate oxidative phosphorylation in mitochondria, it is just one factor that contributes to mitochondrial function. Therefore, we used TMRE and JC-1, vital dyes that have been used previously in neurons and hair cells to study mitochondrial membrane potential–this potential is the main driver of oxidative phosphorylation (*Esterberg et al., 2016*; *Joshi and Bakowska, 2011*). After labeling mature hair cells with TMRE at 6 dpf, we found a significant reduction in TMRE baseline fluorescence in both $ca_V1.3a$ and *otofb* mutant hair cells relative to their respective sibling controls (*Figure 5I–L*). We also labeled mature hair cells using JC-1, a ratiometric indicator of mitochondrial membrane potential. Using this indicator, we also observed a reduction in baseline JC-1 ratio in $ca_V1.3a$ and *otofb* mutant hair cells, although this reduction did not reach statistical significance (*Figure 5—figure supplement 3A-B*). Overall, our TMRE and JC-1 labeling indicates that mitochondrial membrane potential is potentially lower in hair cells with impaired neurotransmission. A lower mitochondrial potential could explain the reduced oxidative stress detected in both $ca_V1.3a$ and *otofb* mutant hair cells.

## $Ca_V1.3a$ and *otofb* mutants exhibit reduced mitochondrial oxidation over time

Our TMRE measurements reflect mitochondrial activity at a single moment in time (*Figure 5I–L*). In contrast our CellROX Orange and MitoSOX Red measurements likely reflect ROS that have accrued over time (*Figure 5A–H*). One drawback to using these single wavelength vital dyes (TMRE, CellROX and MitoSOX) is that they rely on comparable dye uptake between our mutants and controls. Although we observed normal Neo-TR entry into hair cells in our mutants, it is possible that vital dye entry is impaired. Therefore, we examined how mitochondrial activity relates to ROS production over time using a genetically encoded indicator of oxidative stress, MitoTimer. Using a stable transgenic line expressing MitoTimer ensures that the indicator is present at comparable levels in both mutants and controls. MitoTimer localizes to mitochondria and exhibits an oxidation-dependent shift in fluorescence signal from green to red. MitoTimer has previously been used in lateral-line hair cells to detect the accumulation of oxidative stress (*Hernandez et al., 2013*; *Pickett et al., 2018*). For example, Pickett et al. demonstrated that historically older hair cells exhibit a higher red to green fluorescence ratio and are more susceptible to neomycin-induced cell death.

We used MitoTimer to measure differences in mitochondrial oxidation with age in hair cells of $ca_V1.3a$ or *otofb* mutants. The posterior lateral-line is still forming in larvae at 2–3 dpf; at these ages hair cells are newly formed and immature (*Kindt et al., 2012*). In contrast, at 5–6 dpf the lateral-line system is functional, and the majority of hair cells are fully mature. We measured the MitoTimer ratio (red to green) in younger hair cells in larvae at 3 dpf and in older hair cells in larvae at 5 dpf and 6 dpf. Similar to previous work, we observed a significant increase in the Mitotimer ratio, consistent with a buildup of mitochondrial oxidation, in control hair cells over time (*Figure 6D-E*). In addition, we found that the MitoTimer ratio was significantly reduced in older hair cells in both $ca_V1.3a$ and *otofb* mutants at 5 and 6 dpf compared to sibling controls (*Figure 6D–E*). This indicates that in older, mature hair cells lacking $Ca_V1.3$ channels or Otoferlin there is significantly less ROS accumulation.

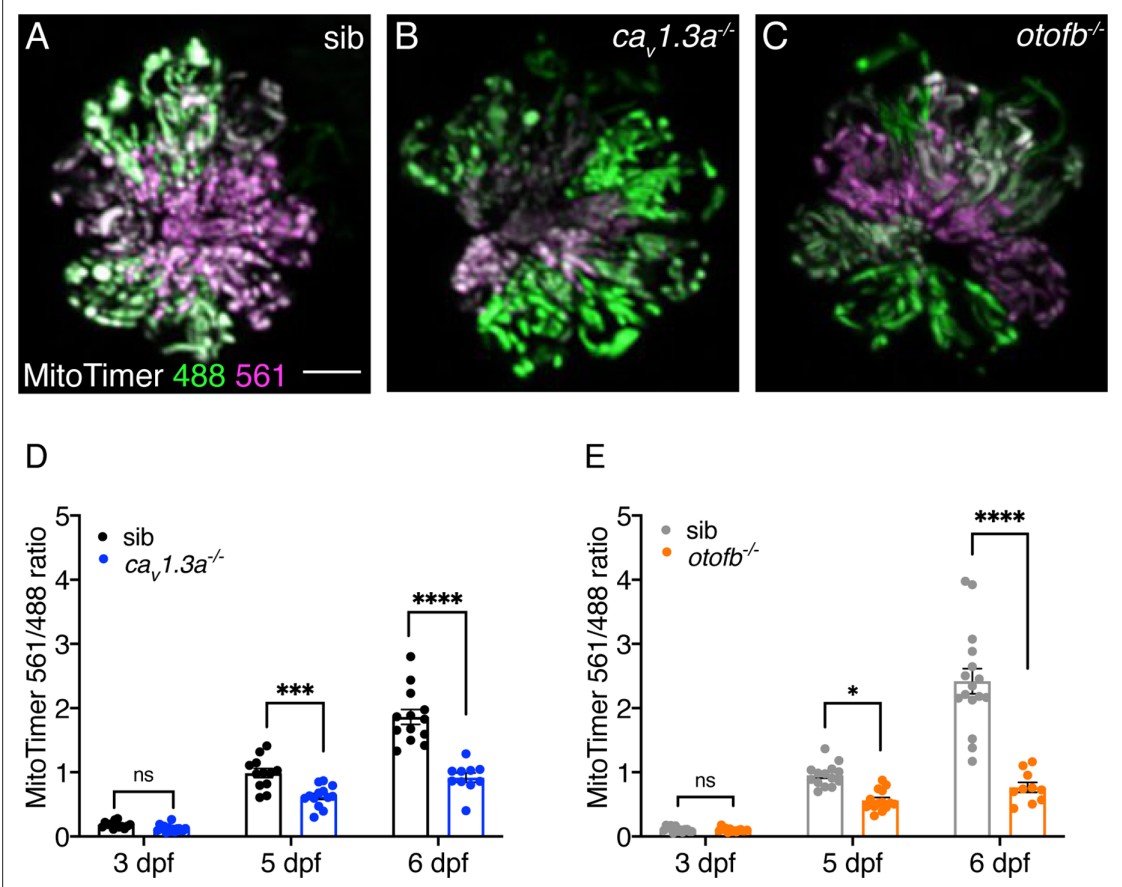

**Figure 6.** Red-shifted oxidized MitoTimer signal is reduced in *ca_v1.3a* and *otofb* mutants. (**A–C**) Hair cells in wildtype (**A**), *cav1.3a⁻/⁻* mutant (**B**), and *otofb⁻/⁻* mutant (**C**) Tg[*myosin6b:mitoTimer*]*^w208* neuromasts at 5 dpf. (**D–E**) Average dot plots of the ratio of red/green MitoTimer fluorescence intensity at 3, 5, and 6 dpf in *cav1.3a⁻/⁻* mutant fish (blue) (**D**) and *otofb⁻/⁻* mutant fish (orange) (**E**) and respective siblings (black, gray) show that mutants exhibit reduced mitochondrial oxidation over time. Each dot in D and E represents one neuromast. A minimum of 3 animals and 10 neuromasts were examined per treatment group. Error bars: SEM. For comparisons, a two-way ANOVA with a Sidak's correction for multiple comparisons was used in D-E. * $p < 0.05$, *** $p < 0.001$, **** $p < 0.0001$. Scale bar = 5 µm.

The online version of this article includes the following source data for figure 6:

**Source data 1.** Mean numbers and statistics for Mitotimer labeling.

Overall, examining mitochondrial oxidation in younger and older hair cells revealed additional insights into the relationship between neurotransmission and mitochondrial oxidation. Our MitoTimer measurements also show that in controls, hair cells accumulate significant mitochondrial oxidation as they mature and age. Importantly, hair cells in both *cav1.3a* and *otofb* mutants both exhibit and maintain a reduction in mitochondrial oxidation as they age. Less mitochondrial oxidation is in line with the lower mitochondrial membrane potential in *cav1.3a* and *otofb* mutants (**Figure 5I–L**). Therefore, a reduction in ROS production in *cav1.3a* and *otofb* mutant hair cells is what augments resistance to cellular stressors such as neomycin.

### The synaptic vesicle cycle modulates hair-cell neomycin susceptibility

Our results indicate that in mature hair cells, either genetic or pharmacological disruption of neuro-transmission results in resistance to ototoxic aminoglycosides. Our manipulations target either Ca_V1.3-channel function (*cav1.3a* mutants or isradipine) or exo- and endo-cytosis (*otofb* mutants or Dynole 34–2). While these manipulations are specific for their respective targets, their functional outcomes overlap. For example, while disruption of exo- and endo-cytosis leaves presynaptic calcium influx intact, disruption of Ca_V1.3-channel function blocks not only presynaptic calcium influx but also downstream exocytosis. Therefore, it is possible that exo- and endo-cytosis rather than presynaptic calcium

influx promotes susceptibility to ototoxins. Alternatively, it is possible that presynaptic calcium influx and exo- and endo-cytosis each independently promote ototoxin susceptibility.

To distinguish between these possibilities, we disrupted Ca$_v$1.3 channel function in *otofb* mutants in order to test if additional protection from neomycin can be obtained by chronically impairing presynaptic calcium influx in these mutants. We treated *otofb* mutants and siblings at 4 dpf with 10 µM isradipine for 24 hr and then challenged larvae at 5 dpf with 100 µM neomycin for 30 min. Quantification at 5 dpf revealed that isradipine treatment did not confer a significant increase in the percentage of surviving hair cells between *otofb* mutants and siblings (*Figure 7A*). In addition, we did not observe any additional protection from neomycin in *otofb* mutants treated with isradipine compared to untreated *otofb* mutants. This data suggests that in mature hair cells, is it the synaptic vesicle cycle rather than presynaptic calcium influx that contributes to oxidative stress and neomycin susceptibility.

## Discussion

In our study we investigated whether the metabolic demands of neurotransmission contribute to the susceptibility of hair cells to ototoxic aminoglycosides. We targeted two critical components of hair-cell neurotransmission: presynaptic calcium influx and the synaptic vesicle cycle. Overall, we found that chronic but not transient disruption of neurotransmission can partially protect hair cells from aminoglycosides. Furthermore, our results show that, over time, neurotransmission leads to an accumulation of metabolic stress by modulating cell physiology at multiple levels: mitochondrial activity, levels of cytotoxic ROS byproducts, and oxidation within mitochondria (*Figure 7B–C*). In line with recent work in neurons, our work suggests that the synaptic vesicle cycle is a strong driver of mitochondrial metabolism and metabolic stress. The accumulation of metabolic stress ultimately renders hair cells more susceptible to cytotoxic insults such as aminoglycoside exposure (*Figure 7C*).

### Neurotransmission and aminoglycoside uptake, clearance, and packaging

A side effect of aminoglycoside treatment in humans is hair-cell loss that results in permanent hearing and balance impairment (*Ariano et al., 2008*; *Bitner-Glindzicz and Rahman, 2007*; *Schacht et al., 2012*). Since this side effect was first discovered, the susceptibility of hair cells to aminoglycosides has been a topic of intense investigation. Hair cells possess a unique pathway of entry for aminoglycosides, namely MET channels (*Alharazneh et al., 2011*). After initial entry through MET channels, aminoglycosides accumulate and remain confined inside hair cells where they can exert their pathological effects. Based on this premise, studies have established that blocking the MET channel is an effective method of protecting hair cells from aminoglycoside-induced cell death (*Alharazneh et al., 2011*; *Kenyon et al., 2021*). However, in our study we found that neurotransmission-deficient hair cells are protected despite largely normal MET channel function as well as normal levels of neomycin accumulation and rates of neomycin entry (*Figures 1 and 3*). Furthermore, our data indicates that the protective effects of blocking neurotransmission are not a result of augmented neomycin clearance (*Figure 3*). Thus, we show that the pathology underlying aminoglycoside exposure extends beyond the kinetics of uptake, accumulation, and clearance.

Consistent with the idea that events downstream of drug entry and accumulation underlie the pathological effects of aminoglycosides, studies have examined how localization and packaging impact aminoglycoside pathology. After entering the hair cell, aminoglycosides rapidly build up in the cytoplasm (*Alharazneh et al., 2011*). In the cytosol, aminoglycosides can interact with the plasma membrane and can accumulate in organelles such as the ER, mitochondria, and lysosomes (*Hailey et al., 2017*; *Hashino et al., 1997*; *Steyger et al., 2003*). The sequestration of aminoglycosides in lysosomes has been of particular interest. For example, work has shown that pharmacologically blocking aminoglycoside uptake into lysosomes exacerbates hair-cell death (*Hailey et al., 2017*). This indicates that lysosomes may play an important role in aminoglycoside degradation. In our study we found that the uptake of neomycin into lysosomes is normal in the absence of neurotransmission (*Figure 3—figure supplement 1*). This indicates that neurotransmission does not overtly modulate the lysosomal sequestration of neomycin and that more rapid or more efficient lysosomal packaging does not explain the protection observed in the absence of neurotransmission.

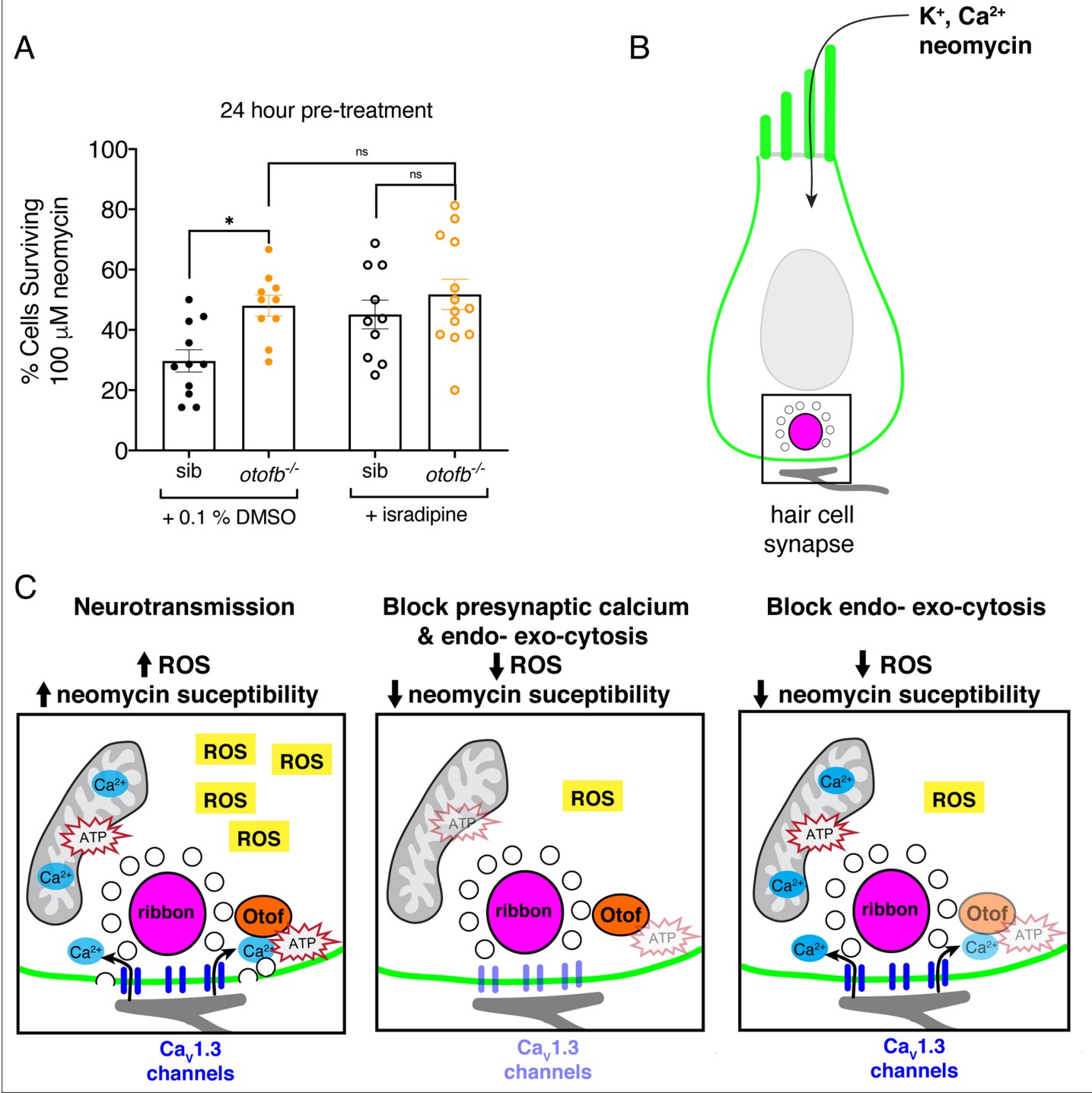

**Figure 7.** The synaptic vesicle cycle is a major contributor to hair-cell oxidative stress. (**A**) Average dot plots show the percentage of cells surviving neomycin treatment per neuromast. A significantly higher percentage of cells survive in *otofb*$^{-/-}$ mutants (orange) compared to siblings (black) when animals are treated with 0.1% DMSO for 24 hr from 4 to 5 dpf. No additional protection is observed in *otofb*$^{-/-}$ mutants (open orange) compared to siblings (open black) when animals are treated with 10 μM isradipine for 24 hr from 4 to 5 dpf prior to neomycin challenge. (**B**) Cartoon schematic of a hair cell with black box around the ribbon (magenta) synapse. (**C**) Close up of synapse demarcated in (**B**). Both presynaptic calcium influx (blue) and otoferlin (orange) function consume ATP (red stars) and stimulate ROS (yellow box) production. Blocking calcium influx inhibits both processes and reduces ROS levels. Blocking the synaptic vesicle cycle alone reduces ROS levels to a similar extent as the calcium channel block. Each dot in A represents one neuromast. A minimum of three animals were examined per treatment group. Error bars: SEM. For comparisons, a two-way ANOVA with a Sidak's correction for multiple comparisons was used in A. * p<0.05.

*Figure 7 continued on next page*

*Figure 7 continued*

The online version of this article includes the following source data for figure 7:

**Source data 1.** Mean numbers and statistics ototoxicity analyses.

Lysosomal packaging is of particular interest because it may explain why neomycin and gentamicin, despite belonging to the same family of antibiotics, exert their effects on very different timescales (*Owens et al., 2009*). For example, in the lateral-line system, gentamicin treatment takes several hours to kill hair cells, as opposed to minutes in the case of neomycin (*Coffin et al., 2013*). It has been shown that, compared to neomycin, gentamicin is much more rapidly packaged into lysosomes following its entry into hair cells (*Hailey et al., 2017*). This rapid packaging is thought to be one reason gentamicin takes longer than neomycin to exert its pathological effects. In addition, studies have shown that these two aminoglycosides may initiate hair-cell death via different pathways (*Coffin et al., 2013*). In our study we find that chronic loss of neurotransmission results in protection from both neomycin and gentamicin (*Figure 2A–E*, *Figure 2—figure supplement 2A-F*), despite different time-courses of pathology, lysosomal packaging and cell-death pathways. This suggests that blocking neurotransmission provides a general form of otoprotection and may broadly protect against differently behaving ototoxic compounds or insults.

## Neurotransmission and cellular metabolism

How could neurotransmission broadly impact susceptibility to a range of ototoxic insults? In order for hair cells to hold up in the face of external insults, in general, it is critical for them to maintain cellular homeostasis. In particular, maintaining metabolic homeostasis is thought to be critical to surviving ototoxin exposure (*Chen et al., 2015*). ROS are byproducts of cellular metabolism and are normally present in low concentrations in all cell types. In hair cells, excess ROS buildup can lead to mitochondrial oxidation and trigger cell-death pathways – especially when hair cells are challenged with an ototoxin (*Huang et al., 2000*). Our results indicate that the energy demands of neurotransmission promote cellular metabolism, which leads to a buildup of ROS and more oxidized mitochondria (*Figures 5–7*). Importantly we show that blocking neurotransmission can lessen metabolic side effects related to these energy demands and protect hair cells from ototoxins.

Like in hair cells, neurotransmission in neurons has also been shown to be energy demanding. Studies in neurons suggest that both the electrical signaling and vesicle release required for neurotransmission require large amounts of ATP (reviewed in: *Harris et al., 2012*). These energy requirements can increase the production of ROS and ultimately result in mitochondrial dysfunction, cellular vulnerability, and neuronal loss (*Singh et al., 2019*). For example, dopaminergic neurons of the substantia nigra depend on the metabolically taxing, ATP-dependent process of calcium clearance for their pace-making function. These neurons have been shown to accumulate ROS to a much greater extent than neighboring neurons. This mitochondrial pathology is thought to lead to cell death and the loss of dopaminergic neurons that underlies Parkinson's Disease (*Dias et al., 2013*). Interestingly, high ATP demands in these dopaminergic neurons are driven by the need to counterbalance the constant L-type calcium channel engagement that sets pacemaker function (*Guzman et al., 2010*). In our work we also find that activity related to L-type calcium channel ($Ca_V1.3$) function increases levels of ROS in hair cells (*Figure 6*). In addition, after chronic pharmacological block of L-type calcium channels, dopamine neurons in the substantia nigra showed significantly lower mitochondrial oxidative stress (*Guzman et al., 2018*). Similarly, in our study we found that chronic block of $Ca_V1.3$ channels reduces mitochondrial activity (*Figure 5*), the amount of ROS (*Figure 5*), and mitochondrial oxidation (*Figure 6*) in hair cells. Thus, the metabolic demands associated with presynaptic calcium flux are high and potentially deleterious to both neuronal and hair-cell homeostasis and health. Furthermore, the pathological effects of these metabolic demands represent a common theme in both hearing loss and neurodegenerative disease (*Singh et al., 2019*; *Wong and Ryan, 2015*).

In addition to presynaptic calcium channel activity, work in neurons has demonstrated that the synaptic vesicle cycle relies heavily on ATP produced by the mitochondria (*Pulido and Ryan, 2021*; *Rangaraju et al., 2014*). Similarly, our work in hair cells also reveals that the metabolic requirements for synaptic vesicle exocytosis – a process downstream of $Ca_V1.3$ channel function – may underlie susceptibility to ototoxins (*Figure 7*). We found that hair cells with intact $Ca_V1.3$ channel function but impaired synaptic vesicle exocytosis (Otofb deficient- and Dynole 34–2 treated-hair cells) were

protected from aminoglycosides (*Figure 2* and *Figure 2—figure supplement 2*). In addition, hair cells with impaired exocytosis also show a reduction in mitochondrial activity (*Figure 5*), oxidation (*Figure 6*) and ROS levels (*Figure 5*). Furthermore, blocking Ca$_V$1.3 channel function in hair cells with impaired exocytosis did not confer any additional protection from neomycin (*Figure 7A*). Together these results indicate that in hair cells, the majority of the ATP required for neurotransmission is used by the synaptic vesicle cycle. In fact, work in neurons has shown that the synaptic vesicle cycle is an underappreciated consumer of ATP (*Pulido and Ryan, 2021*; *Rangaraju et al., 2014*). Within the synaptic vesicle cycle, work in neurons has shown that it is the compensatory endocytosis and synaptic vesicle replenishment following exocytosis that requires significant amounts of ATP. Mechanistically, these ATP demands may arise at least in part from vacuolar-type ATPases (V-ATPases) which reside in synaptic vesicles (*Pulido and Ryan, 2021*). V-ATPases consume ATP in order to help fill vesicles with neurotransmitters such as glutamate and therefore represent a large metabolic burden. Whether the metabolic demands of V-ATPase function are responsible for the metabolic burden and cellular stress associated with hair-cell neurotransmission is not known and awaits future work. Our work in hair cells and complementary work in neurons indicate that the synaptic vesicle cycle represents a source of metabolic stress that can impact cellular health. It is important to understand the specific mechanisms that generate this stress as they represent important targets for therapies to prevent metabolic stress-associated pathologies.

## Long-term consequences of neurotransmission

Many studies on aminoglycoside ototoxicity point to mitochondrial involvement as they are thought to be a target of aminoglycosides (reviewed in: *Foster and Tekin, 2016*). Our study highlights how hair-cell neurotransmission impacts mitochondria – promoting cellular metabolism, leading to a buildup of ROS and more oxidized mitochondria (*Figures 5 and 6*). Under these conditions, when hair cells are challenged with aminoglycosides, mitochondrial pathology linked to neurotransmission and the targeting of aminoglycosides to the mitochondria synergize to initiate hair-cell death.

We also show that as hair cells mature and age, they accumulate more oxidized mitochondria; block of neurotransmission dramatically reduces this oxidation (*Figure 6*). Presbycusis or age-related hearing loss (ARHL) is common in both animal models and humans (*Huang and Tang, 2010*; *Wang and Puel, 2020*; *Wong and Ryan, 2015*). In animal models, ARHL is associated with an accumulation of ROS and mitochondrial damage. Our study suggests that neurotransmission may be a critical link between aging and hair-cell vulnerability. Interestingly, we did observe residual ROS accumulation and mitochondrial oxidation even in hair cells lacking neurotransmission (*Figures 5 and 6*). This indicates that in addition to neurotransmission, there are other factors that promote ROS accumulation and mitochondrial oxidation over time. In the future it will be interesting to use our model to explore what other factors contribute to the accumulation of these pathological effects. By fully understanding the mechanisms underlying ROS-induced mitochondrial oxidation within hair cells, it may be possible to protect hair cells from the damage that they incur over their lifetime.

One therapeutic option to protect hair cells would be to block neurotransmission. But without hair-cell neurotransmission hearing, balance, or lateral-line function are not possible (*Brandt et al., 2003*; *Chatterjee et al., 2015*; *Roux et al., 2006*; *Sidi et al., 2004*). Interestingly, in our previous work we demonstrated that, in the lateral line, the majority of hair cells within neuromast organs are synaptically silent with no detectable evoked presynaptic calcium influx or vesicle fusion (*Figures 1 and 4*; *Zhang et al., 2018*). Given the relationship between hair-cell neurotransmission and metabolic stress, synaptic silencing may be a protective mechanism that occurs in older, more mature hair cells where ROS levels and mitochondrial oxidation are elevated (*Pickett et al., 2018*). This is consistent with our present study where we show that synaptically silent hair cells are relatively older than synaptically active cells and are also more susceptible to neomycin (*Figure 4*, *Figure 4—figure supplement 1*). Therefore, in the lateral line, as hair cells age, synaptic silencing may serve as a mechanism to limit further ROS accumulation and mitochondrial oxidation. In this scenario, historically younger hair cells are tasked to take on neurotransmission, along with the accompanying metabolic burden. A shift of neurotransmission from older to younger hair cells could limit the amount of damage hair cells accumulate and ultimately prevent widespread cell death, especially when neuromasts are faced with insults like neomycin. Additional research is needed to investigate whether synaptic silencing occurs in mammalian hair cells and whether this mechanism could prevent the accumulation of ROS and

mitochondrial oxidation. Such a mechanism would be extremely advantageous as mammalian hair cells do not regenerate and must be maintained over a lifetime.

Overall, our study shows that proper management of metabolic stress including ROS production and mitochondrial oxidation is critical for hair-cell health and resistance to ototoxins. Furthermore, we find that, over time, hair-cell neurotransmission increases ROS production and mitochondrial oxidation resulting in increased vulnerability to aminoglycosides. This aging-associated increase in vulnerability may shed light on presbycusis and the processes that lead to hearing loss with age. Future work will allow us to understand if this vulnerability extends to other classes of insults, such as noise or non-aminoglycoside ototoxic drugs like cisplatin. A better understanding of the pathologies of aging and the effects of ototoxins on hair cells may ultimately contribute to the development of therapies to prevent hearing loss and vestibular dysfunction.

# Methods

**Key resources table**

| Reagent type (species) or resource | Designation | Source or reference | Identifiers | Additional information |
|---|---|---|---|---|
| Strain, strain background (*Danio rerio*) | Tübingen | ZIRC | http://zfin.org/ZDB-GENO-990623-3 | See methods, animals |
| Genetic reagent (*Danio rerio*) | *Tg(myo6b:GCaMP6s-CAAX)$^{idc1}$*; GCaMP6CAAX; | *Jiang et al., 2017* | https://zfin.org/ZDB-ALT-170113-3 | Membrane-localized calcium biosensor |
| Genetic reagent (*Danio rerio*) | *Tg(myo6b:mitoGCaMP3)$^{w119Tg}$*; MitoGCaMP3; | *Esterberg et al., 2014* | https://zfin.org/ZDB-ALT-141008-1 | Mitochondria-localized calcium biosensor |
| Genetic reagent (*Danio rerio*) | *Tg(myo6b:mitoTimer)$^{w208Tg}$*; MitoTimer | *Pickett et al., 2018* | https://zfin.org/ZDB-ALT-190708-1 | Mitochondria-localized ROS biosensor |
| Genetic reagent (*Danio rerio*) | *Tg(myo6b:SypHy)$^{idc6Tg}$*; SypHy | *Zhang et al., 2018* | https://zfin.org/ZDB-ALT-171205-5 | Hair-cell localized vesicle fusion indicator |
| Genetic reagent (*Danio rerio*) | *cav1.3a$^{tn004}$ mutants; gemini; cacna1da* | *Sidi et al., 2004* | https://zfin.org/ZDB-GENE-030616-135 | Mutants lacking functional Ca$_v$1.3 channels |
| Genetic reagent (*Danio rerio*) | *otofb* | This paper | https://zfin.org/ZDB-ALT-211124-4 | Crispr-Cas9 mutant. See Materials and Methods, "Animals" |
| Genetic reagent (*Danio rerio*) | *otofa* | This paper | https://zfin.org/ZDB-ALT-211124-3 | Crispr-Cas9 mutant. See Materials and Methods, "Animals" |
| Sequence-based reagent | *otofa* guide | This paper | PCR primers | 5'- GGGCACCTTCAAACTAGACG(TGG)–3', made by IDT |
| Sequence-based reagent | *otofb* guide | This paper | PCR primers | 5'-GGAGCTCCACTGAGGTGCAGG(TGG)–3', made by IDT |
| Sequence-based reagent | OTOFA FWD | This paper | PCR primers | 5'-ATCAAACCTCCATTGGAAACAG-3', made by Integrated DNA Technologies (IDT) |
| Sequence-based reagent | OTOFA REV | This paper | PCR primers | 5'-CCCATTTGTGATGAAACTGATG-3', made by IDT |
| Sequence-based reagent | OTOFB FWD | This paper | PCR primers | 5'- CTGGTTCATTCGTAGGCTTTCT-3', made by IDT |
| Sequence-based reagent | OTOFB REV | This paper | PCR primers | 5'-TGCTTACATCAGAGATGTTGGG-3', made by IDT |
| Antibody | Otoferlin (mouse monoclonal) | Developmental Studies Hybridoma Bank | RRID:AB_10804296 HCS-1 | Use at 1:1000 |
| Antibody | Alexa Fluor 488 (goat polyclonal) | ThermoFisher | A-11001 | Use at 1:1000 |
| Other | Prolong Gold | ThermoFisher | P10144 | See methods, immunohistochemistry |

*Continued on next page*

*Continued*

| Reagent type (species) or resource | Designation | Source or reference | Identifiers | Additional information |
|---|---|---|---|---|
| Peptide, recombinant protein | α-bungarotoxin | Tocris | 2133 | See methods, paralysis and immobilization |
| Chemical compound, drug | Isradipine; Israd | Sigma-Aldrich | I6658 | See methods, cell-death assays |
| Chemical compound, drug | Tricaine; MESAB | Sigma-Aldrich | A5040 | See methods, paralysis and immobilization |
| Chemical compound, drug | Dynole 34–2 | Tocris Biosciences | 4222 | See methods, cell-death assays |
| Chemical compound, drug | Neomycin; Neo | Sigma-Aldrich | N1142 | See methods, cell-death assays |
| Chemical compound, drug | Gentamicin; Gent | Sigma-Aldrich | G1272 | See methods, cell-death assays |
| Other | FM 4–64 | ThermoFisher | T3166 | See methods, cell-death assays |
| Other | Texas Red-x-succinimidyl ester | ThermoFisher | T20175 | See methods, Neomycin-Texas Red uptake |
| Other | CellROX Orange | ThermoFisher | C10444 | See methods, live imaging |
| Other | MitoSOX Red | ThermoFisher | M36008 | See methods, live imaging |
| Other | TMRE | ThermoFisher | T669 | See methods, live imaging |
| Other | DAPI | ThermoFisher | D1306 | See methods, live imaging |
| Software, algorithm | Prism (v. 8) | Graphpad Software | RRID:SCR_002798; https://www.graphpad.com | See methods, statistics |
| Software, algorithm | Adobe Illustrator | Adobe | RRID:SCR_014198; https://www.adobe.com | See figures |
| Software, algorithm | FIJI is just ImageJ | NIH | RRID:SCR_003070; https://fiji.sc | See methods, image analysis |
| Software, algorithm | PolyPeak Parser | *Hill et al., 2014* | http://yosttools.genetics.utah.edu/PolyPeakParser/ | See methods, animals |
| Software, algorithm | Zen | Zeiss | RRID:SCR_01367; https://www.zeiss.com/microscopy/int/products/microscope-software/zen.html | See methods, immunohistochemistry |
| Software, algorithm | Prairie View | Bruker Corporation | RRID:SCR_017142; https://www.bruker.com/products/fluorescence-microscopes/ultima-multiphoton-microscopy/ultima-in-vitro/overview.html | See methods, functional imaging |

*Continued on next page*

*Continued*

| Reagent type (species) or resource | Designation | Source or reference | Identifiers | Additional information |
|---|---|---|---|---|
| Software, algorithm | G*Power | *Faul et al., 2009* | RRID:SCR_013726; https://doi.org/10.3758/BRM.41.4.1149 http://www.gpower.hhu.de/ | See methods, statistics |
| Software, algorithm | Igor Pro | Wavemetrics | RRID:SCR_000325; http://www.wavemetrics.com/products/igorpro/igorpro.htm | See methods, electrophysiology |
| Software, algorithm | pClamp 10 | Molecular Devices | RRID:SCR_011323; http://www.moleculardevices.com/products/software/pclamp.html | See methods, electrophysiology |

## Animals

Zebrafish (*Danio rerio*) were grown at 28 °C using standard methods. Larvae were raised in E3 embryo medium (5 mM NaCl, 0.17 mM KCl, 0.33 mM $CaCl_2$, and 0.33 mM $MgSO_4$, buffered in HEPES, pH 7.2). Zebrafish work performed at the National Institutes of Health (NIH) was approved by the Animal Use Committee at the NIH under animal study protocol #1362–13. All experiments were performed on larvae aged 3–6 days post fertilization (dpf). Larvae were chosen at random at an age where sex determination is not possible. The following mutant and transgenic lines were used: *Tg(myo6b:GCaMP6s-CAAX)[idc1]* (referred to in this work as memGCaMP6s) (*Jiang et al., 2017*), *Tg(myo6b:SypHy)[idc6]* (*Zhang et al., 2018*), *Tg(myo6b:mitoGCaMP3)[w119]* (*Esterberg et al., 2014*), *Tg(myo6b:mitoTimer)[w208]* (*Pickett et al., 2018*), $ca_v1.3a^{tn004}$ mutants (also known as *gemini or cacna1da*) (*Sidi et al., 2004*). Unless stated otherwise, *Tg(myo6b:memGCaMP6s)[idc1]* larvae were used in all aminoglycoside experiments to assess hair-cell survival.

O*tofa[idc19]* and *otofb[idc20]* mutants were generated in-house using CRISPR-Cas9 technology as previously described (*Varshney et al., 2016*). Exon 11 was targeted in each Otoferlin isoform. Guides for *otofa* and *otofb* are as follows: 5'- GGGCACCTTCAAACTAGACG(TGG)–3' and 5'-GGAGCTCCACTGAGGTGCAGG(TGG)–3'. Founder fish were identified using fragment analysis of fluorescent PCR products. Founder fish containing a 5 bp deletion in *otofa* 5'- CAAACTAC-(TTCAA)-ACGTGGGGA-3' and an 8 bp deletion in *otofb* 5'-AACGAAGG-(CCTCGGGG)-AGGGCGTC-3' were propagated and selected for analysis. Subsequent genotyping was accomplished using standard PCR and sequencing. PolyPeak Parser was used to parse sequences to identify animals with an indel of interest (*Hill et al., 2014*). *Otof* genotypes were confirmed with genotyping after imaging. Primers used for genotyping: *otofa*_FWD 5'-ATCAAACCTCCATTGGAAACAG-3' and *otofa*_REV 5'-CCCATTTGTGATGAAACTGATG-3' and *otofb*_FWD 5'- CTGGTTCATTCGTAGGCTTTCT-3' and *otofb*_REV 5'-TGCTTACATCAGAGATGTTGGG-3'. For ease of identification, *otofa;otofb* double mutants were used for analyses. Loss of *otofb* is sufficient to eliminate all Otoferlin immunolabel in lateral line neuromasts (*Figure 1—figure supplement 1C*). With regard to our neuromast functional imaging, we did not observe any differences between *otofa;otofb* double and *otofb* single mutants.

## Immunohistochemistry and confocal imaging of fixed samples

Immunohistochemistry to label Otoferlin in hair cells was performed on whole zebrafish larvae similar to previous work (*Zhang et al., 2018*). The primary antibody mouse anti-Otoferlin, 1:1000, (HCS-1, Developmental Studies Hybridoma Bank) along with the secondary antibody Alexa 488 1:1000 (A-11001, ThermoFisher Scientific) were used for immunolabel. Larvae were mounted in Prolong gold (P10144, ThermoFisher Scientific). Fixed larvae were imaged on an inverted Zeiss LSM 780 confocal microscope (Carl Zeiss AG) with a 488 nm laser using an 63x1.4 NA oil objective lens. Confocal z-stacks were acquired every 0.3 µm. Imaged were processed using FIJI (NIH).

## Zebrafish immobilization and paralysis for live imaging

Larvae were immobilized for live imaging as described previously (*Lukasz and Kindt, 2018*). Briefly, larvae were mounted on their side on a thin layer of Sylgard atop a chamber with a coverglass bottom in E3 embryo media containing 0.2% MESAB (tricaine; MS-222; ethyl-m-aminobenzoate methane-sulfonate, Western Chemical). Small pins were inserted perpendicularly through the body of the fish behind the ear and into the notochord at the end of the tail. The hearts of pinned larvae were injected with a solution containing α-bungarotoxin (125 μM, 2133, Tocris Biosciences) to paralyze larvae for imaging. After paralysis, larvae were immersed in either CE3 for aminoglycoside experiments (in mM: 14.9 NaCl, 0.503 KCl, 0.986 CaCl$_2$, 0.994 MgSO$_4$, 0.150 KH$_2$PO$_4$, 0.042 Na$_2$HPO$_4$, and 0.714 NaHCO$_3$, pH 7.2) (*Coffin et al., 2009*) or extracellular imaging solution (in mM: 140 NaCl, 2 KCl, 2 CaCl$_2$, 1 MgCl$_2$, and 10 HEPES, pH 7.3, OSM 310+/-10) for functional imaging experiments and electrophysiology.

## Functional imaging of calcium and vesicle fusion

For the imaging of mechanosensation (memGCaMP6s), presynaptic calcium influx (memGCaMP6s), mitochondrial calcium uptake (mitoGCaMP3), and vesicle fusion (SypHy), we used a Bruker Swept-Field confocal system (Bruker Corporation) equipped with a Rolera EM-C2 CCD camera (Qimaging) and a Nikon CFI Fluor 60×1.0 NA water immersion objective (Nikon Instruments, Inc) as described previously (*Zhang et al., 2018*). The system includes a band-pass 488/561 nm filter set (59904-ET, Chroma) and is controlled using Prairie View software (Bruker Corporation). For memGCaMP6s and mitoGCaMP3 imaging we used a piezoelectric motor (PICMA P-882.11–888.11 series, PI Instruments) attached to the objective to acquire rapid five-plane Z-stacks. Z-stacks were acquired with a 1 μm interval to image mechanosensation-dependent calcium flux and a 2 μm interval to image presynaptic and mitochondrial calcium flux. Images were acquired using a 70 μm slit at a 50 Hz frame rate with a resulting a 10 Hz volume rate. Z-stacks were average projected into a single plane for analysis. For SypHy imaging, 3 separate synaptic planes were acquired 2 μm apart at a 50 Hz frame for each neuromast.

A fluid-jet was used to stimulate apical bundles to evoke vesicle fusion as well as calcium-dependent mechanosensation, presynaptic influx, and mitochondrial uptake. Fluid-jet stimulation was described in detail previously (*Lukasz and Kindt, 2018*). Briefly, a fluid-filled glass capillary (part # B150-86-10, Sutter Instruments) was used to deliver anterior and posterior directed fluid flow. Flow was triggered from the capillary using a pressure clamp system (HSPC-2-SB and PV-Pump, ALA Scientific Instruments) controlled by Prairie View software to coordinate imaging with fluid stimulation. A 500 ms or 200 ms step (mechanosensation and presynaptic calcium), 2 s 5 Hz alternating anterior and posterior (vesicle fusion), or 4 s step (mitochondrial calcium) fluid-jet stimulation was used to deflect the apical bundles of anterior-posterior responding posterior lateral line neuromasts.

For Dynole 34–2 pharmacology, calcium-dependent mechanosensation or presynaptic influx was measured in 0.1% DMSO during a 500 ms step stimulus. Then the solution was exchanged with one containing 2.5 μM Dynole 34–2 in 0.1% DMSO or 0.1% DMSO alone. Larvae were incubated in the new solution for 10 min. After this incubation period evoked calcium-dependent mechanosensation or presynaptic influx were reassessed.

To visualize changes in GCaMP6s, mitoGCaMP3, and SypHy signals, raw images were processed using a custom program with a user-friendly GUI interface in MATLAB R2014 (Mathworks) (*Zhang et al., 2018*). This analysis has been described in detail (*Lukasz and Kindt, 2018*). The first 10 time-points from each acquisition were removed to eliminate the initial photobleaching. Then the raw images were registered in X-Y to eliminate movement artifacts. A pre-stimulus reference image was used to generate a baseline image (F$_0$). Then the baseline image (F$_0$) was subtracted from each image acquired to generate a series of images that represent the relative change in fluorescence signal from baseline or ΔF. The ΔF signals were temporally binned (every 0.5 s), scaled and encoded by color maps, with darker colors (orange, red) indicating an increase in signal intensity. The color maps were then superimposed into the baseline (F$_0$) grayscale images in order to visualize the spatial fluorescence intensity changes in hair cells during stimulation.

To quantify the fluorescence intensity changes from our GCaMP6s, mitoGCaMP3, and SypHy functional imaging, registered images were loaded into FIJI (*Schindelin et al., 2012*). In FIJI we used the Time Series Analyzer V3 plugin to create circular ROIs. For presynaptic GCaMP6s, mitoGCaMP3, and SypHy measurements, a circular region of interest (ROI) with a diameter 3.3 μm (~12 pixels

with 268 nm per pixel) was placed at the base of each hair cell within a neuromast. For hair-bundle GCaMP6s measurements, a circular ROI with a 1.7 μm diameter (~6 pixels with 268 nm per pixel) was placed on the center of an individual bundle. We then measured and plotted change in the mean intensity ($\Delta F/F_0$) within the region during the recording period. The mean intensity within each ROI was computed for each cell. The signal magnitude, defined as the peak value of intensity change upon stimulation was determined for all ROIs. In addition, the duration (time from stimulus onset to peak), and the slope (from stimulus onset to peak) was measured. After the stimulus peak, the half-life of the GCaMP6s signal was fitted using an exponential one-phase decay. To quantify the resting or baseline GCaMP6s in hair bundles or presynaptic compartment, the mean intensity during the pre-stimulus acquisition (2 s interval) was determined. For GCaMP6s hair-bundle measurements, GCaMP6s signals from all hair bundles were averaged to give a value for each neuromast. For GCaMP6s, mitoGCaMP3, and SypHy, and presynaptic measurements, only the signals from synaptically active hair cells were averaged to give a value for each neuromast (*Zhang et al., 2018*). For mutants without GCaMP6s, mitoGCaMP3, or SypHy signals, a comparable number of cells were chosen at random for analysis.

## Afferent neuron electrophysiology

Postsynaptic currents from afferent cell bodies of the posterior lateral-line ganglion (pLLg) were recorded as described previously in zebrafish (*Trapani and Nicolson, 2011*). The pLLg was visualized using an Olympus BX51WI fixed stage microscope equipped with a LumPlanFl/IR 60 X/0.90 W water dipping objective (N2667800, Olympus). To record spontaneous postsynaptic currents, borosilicate glass pipettes were prepared with a long taper and resistances between 5 and 10 MΩ (P-97, Sutter Instruments). A Digidata 1440 A data acquisition board, Axopatch 200B amplifier, and pClamp 10 software (Molecular Devices, LLC) were used to collect signals. A loose-patch configuration with seal resistances ranging from 20 to 80 MΩ was used in combination with voltage-clamp mode, and signals were sampled at 50 μs/pt and filtered at 1 kHz. Cell bodies were selected at random because mutants lack evoked spikes and therefore it is impossible to identify specific neuromasts innervated by a given neuron. Igor Pro (Wavemetrics) was used to analyze afferent electrophysiology recordings. The number of spontaneous events per neuron per minute within a 5-min recording window was quantified.

For the Dynole 34–2 assay, spontaneous baseline afferent signals were recorded from a single cell body from a 4–6 dpf larva for a minimum of 5 min. Then a microloader pipette tip (E5242956003, Eppendorf) was used to carefully deliver 10 μL of 125 μM Dynole 34–2 solution (for a final concentration of 2.5 μM Dynole 34–2 in 0.1% DMSO or 0.1% DMSO alone) over 5 s to avoid disrupting the recording. Recordings were continued for another 10 min to determine if the treatment had an effect on spontaneous afferent spike rate. The number of spontaneous events per neuron within the 5-min pre-treatment window was quantified to obtain a baseline before treatment, and the number of spontaneous events per minute the 10-min window after treatment was quantified for post-treatment quantification.

## Neomycin and gentamicin hair-cell death assays

*Tg(myo6b:memGCaMP6s)$^{idc1}$* larvae were immobilized for imaging as described above. Larvae were then incubated in neomycin sulfate solution (75, 100, or 200 μM in CE3 media) for 30 min. After neomycin treatment, larvae were washed one time in CE3 media, followed by a 30 -s incubation with the vital dye FM 4–64 (2 μM, T3166, ThermoFisher), and three more washes in CE3 media. Next, fluorescent, two-color 488 and 561 nm Z-stacks were taken at 1.5 μm intervals using a Nikon A1 upright laser-scanning confocal microscope using a 60×1.0 NA water objective (Nikon Instruments, Inc). The number of surviving cells was identified primarily by counting intact GCaMP6s positive cells that also co-labeled with FM 4–64, a vital dye which disappears upon membrane fragmentation. Any very young or immature GCaMP6s-positive cells without visible hair bundles were excluded from analysis. The number of surviving cells was divided by total cells present before treatment to obtain % cells surviving neomycin treatment for a given concentration for each neuromast.

For the short-term isradipine and Dynole 34–2 assays, larvae were immobilized for imaging then pretreated for 10 min with 10 μM isradipine (I6658, Sigma-Aldrich) or 2.5 μM Dynole 34–2 (4222, Tocris Biosciences) in CE3 media before being co-incubated in isradipine or Dynole 34–2 and 100 μM neomycin sulfate solution in CE3 media for 30 min. This last incubation was followed by a single wash

with CE3 media, incubation for 30 s in 2 µM FM 4–64 in CE3, and two more washes in CE3 followed immediately by imaging. For the long-term isradipine and Dynole 34–2 assays, free swimming 3 dpf or 4 dpf larvae were incubated in 10 µM isradipine or 2.5 µM Dynole 34–2 for 48 or 24 hr, respectively. At 5 dpf larvae were washed 3 X in CE3 before treatment with 100 µM neomycin for 30 min. After short- or long-term drug incubation and neomycin application, neuromasts were imaged and quantified as described above.

For the gentamicin assays, 5 dpf *Tg(myo6b:memGCaMP6s)$^{idc1}$* larvae were incubated in gentamicin sulfate (G1272, Sigma-Aldrich) solution (200 µM in CE3 media) for 2 hr before being washed 3 X in CE3 media and returned to E3 media for 24 hr. After the 24 hr recovery, the 6 dpf larvae were then incubated in 2 µM FM 4–64 for 30 s, followed by two washes in CE3 before imaging. Neuromasts were imaged as described for the neomycin cell death assay. GCaMP6s-positive cell bodies labeled with FM 4–64 were counted for both treated and untreated larvae. The number of surviving cells in the treated group was normalized by the number of cells in the untreated group for each genotype to obtain % cells surviving gentamicin treatment for each neuromast.

## Neomycin-Texas Red uptake and clearance assays

To monitor neomycin uptake and clearance, neomycin sulfate (N1142, Sigma-Aldrich) was conjugated to Texas Red-X-succinimidyl ester (T20175, ThermoFisher) as described previously (*Stawicki et al., 2014*). For imaging of the temporal components of neomycin-Texas Red (Neo-TR) uptake and clearance, an A1 Nikon upright laser-scanning confocal microscope with a 60×1.0 NA water objective was used to acquire images Z-stacks every 1.5 µm. For Neo-TR uptake, *Tg(myo6b:memGCaMP6s)$^{idc1}$* larvae were first imaged using a 488 nm laser in 1 mL CE3 to locate neuromasts then 1 mL of 50 µM Neo-TR was added to the solution for a final concentration of 25 µM. Two-color 488 and 561 nm imaging was begun immediately following the addition of Neo-TR with Z-stacks collected every 1 min for 10 min. Immediately following the collection of the final Z-stack for the uptake portion of imaging, the Neo-TR solution was removed, and larvae were washed 3 X with 1 mL of CE3. After the washout Z-stacks were collected every 90 s for 30 min to track Neo-TR clearance. After imaging, Neo-TR intensity was quantified using FIJI. For both uptake and clearance experiments, the Neo-TR channel was corrected for drift in FIJI. Then Z-stacks were max-projected. Neo-TR and GCaMP6s channels were max-projected, and the Neo-TR channel was background subtracted (rolling ball radius 50 pixels). The GCaMP6s channel was then subjected to mean automatic thresholding to generate a mask. This mask was applied to the maximum projected Neo-TR image and the multi-measure tool in FIJI was used to extract the average Texas Red intensity at each time point. The change in Neo-TR fluorescence intensity (ΔF) was plotted over time.

## Live imaging of CellROX Orange, MitoSOX Red, TMRE, JC-1, MitoTimer, and DAPI

For baseline measurements, CellROX Orange (C10444, ThermoFisher), MitoSOX Red (M36008, ThermoFisher), TMRE (T669, ThermoFisher) and JC-1 (T3168, ThermoFischer) dyes were applied at 10 µM, 5 µM, 10 nM, and 1.5 µM respectively, in CE3. *Tg(myo6b:memGCaMP6s)$^{idc1}$* larvae were incubated in CellROX Orange or TMRE for 30 min or MitoSOX for 15 min in darkness. Nontransgenic larvae were incubated in JC-1 for 5 min in darkness. After dye incubation larvae were washed 3 X with CE3, and fluorescent 2-color 488 and 561 nm Z-stacks were taken at 2–2.5 µm intervals using a Nikon A1 upright laser-scanning confocal microscope using a 60×1.0 NA water objective. Laser and Z-stack settings were kept constant across each experiment. After imaging, the two most central slices encompassing the majority of the hair cell bodies were max-projected using FIJI. Mean automatic thresholding applied to the green channel (GCaMP6s) was used to create a mask encompassing all mature neuromast hair cells. This mask was used to create an ROI to measure the average intensity in the red channel (CellROX Orange, MitoSOX Red, or TMRE). For JC-1 measurements the green channel was used to generate a threshold and mask to quantify the intensity in both the green and red channels.

*Tg(myo6b:mitoTimer)$^{w208}$* larvae were immobilized as described above and imaged at 3, 5, or 6 dpf using a Nikon A1 upright laser-scanning confocal microscope. Z-stacks of the 488 and 561 nm channels were collected at 2 µm intervals through the neuromast from apex to base. After imaging, max-projections of the green and red channel Z-stacks through the entire neuromast were generated using FIJI. Both channels were background subtracted with a mean rolling ball radius of 50 pixels.

Mean automatic thresholding was applied to the green channel to generate a mask encompassing all MitoTimer-expressing neuromast hair cells. This mask was used to create an ROI to measure the average intensity of the green and red channel, then the ratio of the average fluorescence intensity in the red divided by the green channel was calculated for each neuromast.

DAPI labeling and kinocilial height measurements were used to differentiate between older and younger cells within a given neuromast at 5 dpf. To measure the height of the tallest part of the hair bundle, the kinocilium, a transmission PMT on a laser-scanning Nikon A1 confocal microscope was used to acquired Z-stacks every 0.5 µm. For DAPI labeling, 14 µM DAPI nuclear dye (D1306, ThermoFisher) in CE3 was applied to *Tg(myo6b:memGCaMP6s)$^{idc1}$* larvae for 30 s at 3 dpf. Larvae were then washed 3 X in CE3 before being returned to E3 solution. After labeling larvae were imaged in 405 and 488 nm channels using a Nikon A1 confocal microscope to identify DAPI-positive and DAPI-negative cells at 3 dpf or placed directly in an incubator at 28 °C. At 5 dpf, DAPI-labeled neuromasts were imaged to identify DAPI-positive and DAPI-negative cells. For calcium imaging experiments, after identifying DAPI-positive cells, presynaptic calcium imaging was performed using 2 s anterior and posterior step fluid-jet stimuli on the Bruker Swept-Field confocal microscope as described above.

## Statistics

All data shown are mean ± standard error of the mean (SEM). All experiments were replicated at least twice on two independent days from different clutches. All replicates were biological-distinct animals and cells. Wildtype animals were selected at random for drug pre-treatments. When possible, control animals and drug treated or mutant animals were treated and examined simultaneously (same imaging chamber). Datasets were excluded only when control experiments failed. For experiments relying on single wavelength vital dyes (MitoSOX, CellROX and TMRE) trends were confirmed in a minimum of 3 independent experiments. In all datasets dot plots represent the 'n'. N represents either the number of neuromasts, hair cells or the number of afferent neurons as stated in the legends. For all experiments a minimum of 3 animals and 6 neuromasts or afferent neurons were examined. Exact numbers are listed in the Source Data Files for each figure. Power analyses to determine appropriate sample sizes were performed using G*Power 3.1 (*Faul et al., 2009*). Sample sizes with adequate size were used to avoid Type 2 error. All statistical analysis was performed using Prism 8.0 software (GraphPad). A D'Agostino-Pearson normality test was used to test for normal distributions and F test was used to compare variances. To test for statistical significance between two samples, either paired or unpaired t-tests (normally distributed data), or Wilcoxon or Mann-Whitney tests (not normally distributed data) were used. For multiple comparisons a one-way ANOVA (normally distributed) with a Tukey or Dunnett's correction, a two-way AVOVA (normally distributed data) with a Sidak's correction or a Kruskal-Wallis test (not normally distributed data) with a Dunn's correction was used as appropriate. A p-value less than 0.05 was considered significant. Raw data represented in dot plots, along with the mean, SEM, statistical test used, and exact p values are lists in the Source Data files.

## Acknowledgements

We thank Dave Raible for providing the *myo6b:mitoTimer and myo6:mitoGaMP3* transgenic lines. We thank Brynnae Harrod for her contribution to early measurements of CellROX labeling. We also thank Allison Coffin for providing a protocol to create conjugated Neo-TR. Lastly, we thank Katie Drerup, Sian Kitcher and Elizabeth Cebul for thoughtful comments on the manuscript. This work was supported by a National Institute on Deafness and Other Communication Disorders (NIDCD) Intramural Research Program Grant 1ZIADC000085-01 (KSK).

## Additional information

### Funding

| Funder | Grant reference number | Author |
| --- | --- | --- |
| National Institute on Deafness and Other Communication Disorders | 1ZIADC000085-01 | Daria Lukasz<br>Alisha Beirl<br>Katie Kindt |

| Funder | Grant reference number | Author |
|---|---|---|

The funders had no role in study design, data collection and interpretation, or the decision to submit the work for publication.

## Author contributions

Daria Lukasz, Conceptualization, Data curation, Formal analysis, Investigation, Writing - original draft, Writing - review and editing; Alisha Beirl, Data curation, Investigation, Methodology; Katie Kindt, Conceptualization, Data curation, Formal analysis, Supervision, Funding acquisition, Investigation, Writing - original draft

## Author ORCIDs

Daria Lukasz ⓘ http://orcid.org/0000-0002-8220-7712
Katie Kindt ⓘ http://orcid.org/0000-0002-1065-8215

## Ethics

Zebrafish work performed at the National Institutes of Health (NIH) was approved by the Animal Use Committee at the NIH under animal study protocol #1362-13.

## Decision letter and Author response

Decision letter https://doi.org/10.7554/eLife.77775.sa1
Author response https://doi.org/10.7554/eLife.77775.sa2

# Additional files

## Supplementary files

• Transparent reporting form

## Data availability

All summary data (Figures 1-7 and all supplements) generated or analyzed in this study are included in the Source Data files.

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
