## [Editor Report]

Lukasz and colleagues report important new results revealing how changes in zebrafish lateral line hair cell synaptic activity results in increased vulnerability to ototoxic insult. The authors provide convincing evidence for neurotransmitter release altering susceptibility to aminoglycoside exposure through experiments examining mutants where synaptic release is disrupted. Changes in synaptic activity are accompanied by modest but significant changes in mitochondrial activity, consistent with previous studies revealing that mitochondrial changes impact hair cell susceptibility to damage. This work will inform future studies on how accumulating damage contributes to hair cell damage and ultimately hearing and balance disorders.

---

## [Decision Letter]

**Decision letter after peer review:**

Thank you for submitting your article "Chronic neurotransmission increases the susceptibility of lateral-line hair cells to ototoxic insults" for consideration by *eLife*. Your article has been reviewed by 2 peer reviewers, and the evaluation has been overseen by a Reviewing Editor and Didier Stainier as the Senior Editor. The following individual involved in review of your submission has agreed to reveal their identity: Alan Cheng (Reviewer #1).

Essential revisions:

1. The study uses two mutant zebrafish line and claims that they have "normal mechanotransduction as defined by GCAMP6s fluorescence and neo-TR uptake". Since aminoglycoside entry and toxicity can be altered by MET channel properties (O'Sullivan et al., PNAS 2020, Kenyon et al., JCI insights 2021, Hailey et al., JCI 2016), whether such properties remain completely normal in these mutant hair cells remain to be determined. The conclusion that MET is not involved seems unsubstantiated especially when the degree of toxicity reduction is rather small. Moreover, Trapani and Nicolson 2011 report that CaV1.3 mutants have reduced microphonics consistent with altered MET. It would be critical to establish that otof mutants have normal microphonics for robust conclusions to be drawn.

2. One would expect the intracellular calcium level as a whole pool be affected in the Cav1.3 mutants. Although the authors show there is no difference between those mutant and control hair cells in terms of calcium responses, are baseline levels lower?

3. An alternative explanation for why mutant hair cells are more resistant is that cells are actually younger. Mutants may show increased turnover of hair cells. The authors could address this point by using the DAPI dye labeling assay.

4. Cav1.3 KO hair cells in mice appear less differentiated suggesting abnormal development (Eckrich et al., Frontiers Cell Neuro 2019). This should be considered in discussing potential alternatives.

5. The authors make a point to show the differences in activity between "old and young" hair cells, and differences in survival between active and inactive cells. This leads to the question of whether the surviving hair cells are mostly young or old? This could be tested directly.

6. Experiments to test that Dynol 34-2 specifically affects synaptic signaling appear underpowered (Figure 2-2). Reviewer 1 notes that different tests are used for the different analyses even though both experiments compare pre and post-treatment. Additional experiments should be considered to support conclusions.

7. The majority of measures of mitochondria oxidation and activity use the fluorescence levels of a single colored dye, TMRE. If for some reason these dyes were not able to get into the hair cells or mitochondria of the mutant animals this would confound their results. The concern about the single color dyes could potentially be addressed by using a ratiometric dye like JC-1 in place of TMRE.

8. There are several issues with regards to statistical analysis that need to be addressed, although re-analysis is unlikely to alter conclusions.

Figure 2: D, E should be analyzed as a two-way ANOVA for dose and genotype.

Figure 3 C,D,E,F should be analyzed as a two-way ANOVA for time and genotype.

Figure 4 B should be analyzed as a two-way ANOVA for dose and active state.

Figure 5B: was an outlier test performed? 3 points look like potential outliers.

Figure 5 B,D, Figure 6 B,C inconsistent use of Mann-Whitney vs T-test. Analysis should be consistent or explained.

Figure 7D,E should be analyzed as a two-way ANOVA for time and genotype.

Figure 8A should be analyzed as a two-way ANOVA for drug treatment and genotype.

*Reviewer #1 (Recommendations for the authors):*

1) The study uses two mutant zebrafish line and claims that they have "normal mechanotransduction as defined by GCAMP6s fluorescence and neo-TR uptake". First one would expect the intracellular calcium level as a whole pool be affected in the Cav3/1 mutants, can the author explain why there is no difference between those mutant and control hair cells? Second, aminoglycoside entry and toxicity can be altered by MET channel properties (O'Sullivan et al., PNAS 2020, Kenyon et al., JCI insights 2021), whether such properties remain completely normal in these mutant hair cells remain to be determined. The conclusion that MET is not involved seems unsubstantiated especially when the degree if toxicity reduction is rather small. Third Cav1.3 KO hair cells in mice appear less differentiated suggesting abnormal development (Eckrich et al., Frontiers Cell Neuro 2019). While ephys experiments is not required, revising the interpretation and conclusion seems appropriate.

2) The authors looked at neomycin uptake/clearance using Neomycin-TR and claim that there were no differences in lysosomal loading in Cav1.3a or Otof mutants by the presence of "bright puncta" (Figure 3-S1). The use of a lysosomal marker, such as LAMP1 for example, should be done to support their claim. Co-labelling for Neomycin-TR and a lysosomal marker would support the claims made. As the manuscript stands, the claim that neomycin protection observed in Cav1.3 or otofb mutants is not due to trafficking into lysosomes is not supported.

3) In Figure 4, the authors show that synaptically active cells are young and resistant to neomycin, and that most cells that survive neomycin are "active". This contrasts with their overall findings that lessening metabolic demands, reduces ROS, leading to partial protection from ototoxic drugs. The confounding statements leads to overall confusion in the message of the protective mechanism. Furthermore, since the authors make a point to show the differences between "old and young" hair cells, this leads to the question of whether the surviving hair cells are mostly young or old from Figure 1? It would be beneficial to quantify this in the earlier figures to separate those populations. If would be ideal if there are alternative markers for older and younger cells than DAPI.

*Reviewer #2 (Recommendations for the authors):*

The concern about the single color dyes could potentially be addressed by using a ratiometric dye like JC-1 in place of TMRE or using a chronic treatment of one of the synaptic drugs in place of the mutants and then washing it out before dye treatment. At the very least the caveat of potential drug uptake confounds should be discussed in the manuscript.

---

## [Author Response]

Essential revisions:1. The study uses two mutant zebrafish line and claims that they have "normal mechanotransduction as defined by GCAMP6s fluorescence and neo-TR uptake". Since aminoglycoside entry and toxicity can be altered by MET channel properties (O'Sullivan et al., PNAS 2020, Kenyon et al., JCI insights 2021, Hailey et al., JCI 2016), whether such properties remain completely normal in these mutant hair cells remain to be determined. The conclusion that MET is not involved seems unsubstantiated especially when the degree of toxicity reduction is rather small. Moreover, Trapani and Nicolson 2011 report that CaV1.3 mutants have reduced microphonics consistent with altered MET. It would be critical to establish that otof mutants have normal microphonics for robust conclusions to be drawn.

Good suggestion. Although this is a great experiment, our group does not have the knowledge and experience needed to make quantitative measurements of microphonics in a timely manner. In addition, the majority of studies that have examined microphonic potentials from neuromasts, although powerful, are not quantitative but rather qualitative. In our revision we made a serious attempt to reexamine MET the way we do best, using GCaMP. We examined GCaMP6s-dependent MET responses more carefully in both mutants. For our reanalysis we initiated a completely new set of experiments and reexamined MET responses during 2 stimulus durations (200 and 500ms) instead of 1 duration (500ms). We also examined additional features of the calcium-dependent MET response beyond response magnitude/MAX. This includes resting calcium in the hair bundles, as well as the duration, slope, and response recovery (return to baseline). Our analysis revealed no differences across the board for otof mutants compared to controls. Similar to our previous results we observed no difference in ca_V_1.3 mutants with regard to the magnitude of the MET-response during a 200- or 500-ms stimulus. But we did observe some differences in MET-responses of ca_V_1.3 mutants compared to controls. Ca_V_1.3 mutants have higher resting calcium levels in hair bundles (p = 0.08) and the recovery of the MET response (return to baseline) is significantly faster compared to controls. These additional properties are now compiled in Figure 1-S1. In addition, these new results are included in the Results section.

We also state, “Because the resting calcium levels in hair bundles are higher in ca_V_1.3a mutants and the recovery of the GCaMP6s responses is faster, it is possible that ca_V_1.3a mutants have subtle defects in mechanotransduction. In the future electrophysiological approaches are needed to assess mechanotransduction more carefully in both ca_V_1.3a and otofb mutants. Overall, our GCaMP6s measurements indicate that both Ca_v_1.3a- and Otofb-deficient hair cells exhibit largely normal evoked mechanotransduction.”

2. One would expect the intracellular calcium level as a whole pool be affected in the Cav1.3 mutants. Although the authors show there is no difference between those mutant and control hair cells in terms of calcium responses, are baseline levels lower?

We have now used our GCaMP6s transgenic line to investigate not only evoked MET responses but also resting calcium levels in both the hair bundles and at the presynapse in ca_V_1.3a and otof mutants. We find that in otof mutant resting calcium levels in the hair bundles and at the presynapse are unaltered compared to controls. This is in line with data showing normal evoked MET and presynaptic calcium responses in otof mutants. In previous work we have shown that block of Ca_V_1.3 channels with isradipine lowers resting cytosolic and mitochondrial calcium levels https://elifesciences.org/articles/48914 (see Figure 6A-F in this older paper). We now show that Ca_V_1.3a-deficient hair cells have significantly lower presynaptic calcium levels (Figure 1—figure supplement 1I). In addition (as mentioned above) calcium levels in the hair bundles are higher in ca_V_1.3a mutants compared to controls (p = 0.08; Figure 1—figure supplement 1A). So yes! many pools of calcium are disrupted in ca_V_1.3a mutants. But otof mutants have no detectable changes with regard to baseline or evoked calcium responses. Overall, it was using these 2 mutants in combination that helped drive our conclusion linking exocytosis rather than calcium handling with protection from neomycin.

3. An alternative explanation for why mutant hair cells are more resistant is that cells are actually younger. Mutants may show increased turnover of hair cells. The authors could address this point by using the DAPI dye labeling assay.

Great suggestion. We have added an additional experiment to examine hair cell turnover in ca_V_1.3a and otof mutants compared to controls (See Figure 5—figure supplement 1). Here we used DAPI to label hair cells at day 3 and then examined how many DAPI+ hair cells remained at day 5. We chose this window to assay turnover because it was relevant to the ages used in our experiments. We found that the number of DAPI+ hair cells at day 3 vs day 5 was not significantly reduced in controls or in ca_V_1.3a and otofb mutants (Figure 5—figure supplement 1D, G). This indicates that increased turnover cannot explain the protection ca_V_1.3a and otof mutants. This data is now included in a new Results section “Ca_v_1.3a and otofb mutants show relatively normal maturation and no hair cell loss”

4. Cav1.3 KO hair cells in mice appear less differentiated suggesting abnormal development (Eckrich et al., Frontiers Cell Neuro 2019). This should be considered in discussing potential alternatives.

Good point. It is likely that Ca_V_1.3 could be required for many aspects of hair cell development-as well as cell survival. In the results we now reference the Eckrich citation and others that link hair cell activity to development and cell survival. To try to understand if hair cell development or survival is impacted we have included additional experiments. Overall, our DAPI experiments (see the above point and Figure 5—figure supplement 1D, G) suggest that the same hair cells are alive at 3 and 5 dpf in all genotypes, indicating that there is not a dramatic increase in cell death, implying that cell survival is not affected in the mutants.

In the lateral line work has shown that when mechanotransduction is impaired, fewer hair cells are formed (for example in cdh23 and tmc2 mutants; https://www.nature.com/articles/s41467-017-01604-2#Sec2 (see the supplement of this paper)), linking hair-cell activity to development. Based on this work, we quantified the overall number of hair cells at 3 and 5 dpf in the mutants compared to controls and found no difference (Figure 5—figure supplement S1B, E). Lastly, we examined the height of the tallest part of the hair bundle (the kinocilium) in our mutants compared to controls. Previous work has shown that these measurements can be used to developmentally stage hair cells (https://pubmed.ncbi.nlm.nih.gov/22898777/). Our kinocilial height measurements at 5 dpf were not different in our mutants compared to controls. Overall, both our cell counting and hair bundle measurements point towards relatively normal development in ca_V_1.3a and otof mutants.

This data is now included in a new Results section “Ca_v_1.3a and otofb mutants show relatively normal maturation and no hair cell loss”

5. The authors make a point to show the differences in activity between “old and young” hair cells, and differences in survival between active and inactive cells. This leads to the question of whether the surviving hair cells are mostly young or old? This could be tested directly.

Previous work has demonstrated that historically younger hair cells have less oxidized mitochondria. Furthermore, this work found that hair cells with less oxidized mitochondria are resistant to neomycin (https://elifesciences.org/articles/38062). Other work in the lateral line has shown that lateral-line hair cells are not susceptible to ototoxins until after 4 dpf, suggesting hair-cell age is an important determinant of neomycin susceptibility (https://www.sciencedirect.com/science/article/pii/S0378595503002594#FIGURE 5). In our present work we used a pulse chase of DAPI to identify old versus young hair cells and linked activity to younger cells. Overall, there are a lot of correlative analyses linking younger cells to neomycin resistance.

In our revision, we have attempted to more directly link the age of individual hair cells with susceptibility to the ototoxin neomycin. First, we performed another pulse-chase with DAPI, abelling at 3 dpf, and then examined hair cells at 5 dpf. At 5 dpf we measured the height of the tallest part of the hair bundles, the kinocilium, to assess hair cell maturity in older, DAPI+ cells and younger, DAPI- cells. Here we found that at 5 dpf, all older, DAPI+ hair cells had a kinocilial height of 20µm or greater (Figure 4—figure supplement 1A-C). After using kinocilial measurements to classify old/young cells (young<20µm; old≥20µm), we applied neomycin to hair cells. From this work we found that 66% of younger cells survived, compared to 14.5% of older cells (Figure 4—figure supplement 1D-E’). These experiments strongly indicate that younger cells are significantly more likely to survive. These results are also now included in the Results section.

6. Experiments to test that Dynol 34-2 specifically affects synaptic signaling appear underpowered (Figure 2-2). Reviewer 1 notes that different tests are used for the different analyses even though both experiments compare pre and post-treatment. Additional experiments should be considered to support conclusions.

We have now added additional numbers to support the functional assessment of Dynole 34-2. We added additional electrophysiological recordings of spontaneous afferent spikes before and after Dynole 34-2. This has added power to these analyses (Figure 2—figure supplement 1E-F).

In addition, we redid the calcium imaging experiments to make similar pre- and post-comparisons. Previously, analyses were different between calcium imaging and electrophysiology because in the calcium imaging experiments -in control samples- we observed a reduction in signal magnitude after the repeated recordings needed to capture both the MET- and presynaptic calcium measurements. (This was not the case for our electrophysiology data -no reduction in spikes was observed in our control group (Figure 2—figure supplement 1E)). Because of this reduction in controls we had normalized the calcium imaging data to our mock treated groups. This made the data presentation confusion.

Upon redoing the calcium imaging we performed the MET and presynaptic measurements in independent samples to minimize the number of our recordings. Now there is no difference in our control samples pre- and post-treatment (Figure 2—figure supplement 1A, C). Overall, now the calcium imaging comparisons are more straightforward and in line with our electrophysiology data.

In Figure 2—figure supplement 1 A-E the data follows a normal distribution but in F one of the datasets does not. Because of these differences we used a paired-test (A-E) and Wilcoxon test (F), respectively. Our statistics section now more clearly outlines when different tests are used. In addition, the source data files highlight data distributions and tests used.

7. The majority of measures of mitochondria oxidation and activity use the fluorescence levels of a single colored dye, TMRE. If for some reason these dyes were not able to get into the hair cells or mitochondria of the mutant animals this would confound their results. The concern about the single color dyes could potentially be addressed by using a ratiometric dye like JC-1 in place of TMRE.

It is true that vital dyes relying on a single wavelength are prone to confounds due to variability or inability to permeate and label. With regards to measurements of mitochondrial membrane potential using TMRE, we verified that there are differences between ca_V_1.3 and otof mutants compared to controls in three separate sets of experiments. In our methods section we also now state, “For experiments relying on single wavelength vital dyes (MitoSOX, CellROX and TMRE) differences were confirmed in a minimum of 3 independent experiments.”

We did make a solid effort towards using the ratiometric dye JC-1 to quantify differences in mitochondrial membrane potential in our mutants. In our hands we found this dye to be quite toxic and reduced the incubation time dramatically from previous work. Regardless of these challenges, we examined JC-1 label in our mutants. Using JC-1 we saw a reduction in ca_V_1.3 and otof mutants, but this data did not reach significance (p=0.07 for cav1.3 and p=0.34 for otof). This is now shown in Figure 5—figure supplement 3 and is included in our Results section.

8. There are several issues with regards to statistical analysis that need to be addressed, although re-analysis is unlikely to alter conclusions.Figure 2: D, E should be analyzed as a two-way ANOVA for dose and genotype.Figure 3 C,D,E,F should be analyzed as a two-way ANOVA for time and genotype.Figure 4 B should be analyzed as a two-way ANOVA for dose and active state.Figure 5B: was an outlier test performed? 3 points look like potential outliers.Figure 5 B,D, Figure 6 B,C inconsistent use of Mann-Whitney vs T-test. Analysis should be consistent or explained.Figure 7D,E should be analyzed as a two-way ANOVA for time and genotype.Figure 8A should be analyzed as a two-way ANOVA for drug treatment and genotype.

The statistics above have been updated – all the two-way ANOVAs have been performed as suggested, thank you.

Figure 5B. The dye abelling data is quite variable. Currently we do not feel comfortable doing an outlier analysis to remove datapoints. The overall result was observed in multiple independent experiments. Just one of those experiments is shown in Figure 5B. We also observed a similar amount of variability in all of our experiments, suggesting that the variability may be a part of the biology. Because of the reproducibility of the results we are confident in the data as it stands.

There are what seem to be some inconsistences with the statistics – for example similar datasets using a Mann-Whitney vs a t-test. In part sometime these inconsistences arise because some datasets follow a normal distribution, while other similar datasets may have a condition that leads to a non-normal distribution of the data.

We have done our best to clarify our pipeline in the Statistics section (and in the source data files).

This is what we have added to our methods/statistics section:

“A D’Agostino-Pearson normality test was used to test for normal distributions and F test was used to compare variances. To test for statistical significance between two samples, either paired or unpaired t-tests (normally distributed data), or Wilcoxon or Mann-Whitney tests (not normally distributed data) were used. For multiple comparisons a one-way ANOVA (normally distributed) with a Tukey or Dunnett’s correction, a two-way AVOVA (normally distributed data) with a Sidak’s correction or a Kruskal-Wallis test (not normally distributed data) with a Dunn’s correction was used as appropriate.”

Reviewer #1 (Recommendations for the authors):1) The study uses two mutant zebrafish line and claims that they have “normal mechanotransduction as defined by GCAMP6s fluorescence and neo-TR uptake”. First one would expect the intracellular calcium level as a whole pool be affected in the Cav3/1 mutants, can the author explain why there is no difference between those mutant and control hair cells? Second, aminoglycoside entry and toxicity can be altered by MET channel properties (O’Sullivan et al., PNAS 2020, Kenyon et al., JCI insights 2021), whether such properties remain completely normal in these mutant hair cells remain to be determined. The conclusion that MET is not involved seems unsubstantiated especially when the degree if toxicity reduction is rather small. Third Cav1.3 KO hair cells in mice appear less differentiated suggesting abnormal development (Eckrich et al., Frontiers Cell Neuro 2019). While ephys experiments is not required, revising the interpretation and conclusion seems appropriate.

See the discussion related to MET function above. In addition to the analyses to more carefully examine MET function we have also done our best to soften our statements regarding MET function being ‘normal’ in cav1.3 and otof mutants.

2) The authors looked at neomycin uptake/clearance using Neomycin-TR and claim that there were no differences in lysosomal loading in Cav1.3a or Otof mutants by the presence of "bright puncta" (Figure 3-S1). The use of a lysosomal marker, such as LAMP1 for example, should be done to support their claim. Co-labelling for Neomycin-TR and a lysosomal marker would support the claims made. As the manuscript stands, the claim that neomycin protection observed in Cav1.3 or otofb mutants is not due to trafficking into lysosomes is not supported.

This is a really important point to clarify. A previous study, (https://www.jci.org/articles/view/85052) did an excellent job of characterizing Neo-TR puncta in lateral-line hair cells. For this work they used LysoTracker dye and GFP-Rab7 to show that the majority of Neo-TR puncta are indeed lysosomes. We now reference this work more clearly in our Results section.

3) In Figure 4, the authors show that synaptically active cells are young and resistant to neomycin, and that most cells that survive neomycin are "active". This contrasts with their overall findings that lessening metabolic demands, reduces ROS, leading to partial protection from ototoxic drugs. The confounding statements leads to overall confusion in the message of the protective mechanism. Furthermore, since the authors make a point to show the differences between "old and young" hair cells, this leads to the question of whether the surviving hair cells are mostly young or old from Figure 1? It would be beneficial to quantify this in the earlier figures to separate those populations. If would be ideal if there are alternative markers for older and younger cells than DAPI.

See the response above in the essential revisions #5. For our analyses we have now used both DAPI label and hair bundle height to delineate young and old hair cells (Figure 4—figure supplement v1A-C). We also use hair bundle measurements as a way to directly show that older hair cells in the lateral line are more susceptible to neomycin (Figure 4—figure supplement 1D’E). It is interesting that younger hair cells are more likely to exhibit presynaptic activity than older hair cells. We believe that this activity is common to young, newly formed cells and that the activity ceases as cells age, thus leaving older hair cells both inactive and more susceptible to neomycin due to the fact that they previously experienced significant amounts of presynaptic activity.

Reviewer #2 (Recommendations for the authors):The concern about the single color dyes could potentially be addressed by using a ratiometric dye like JC-1 in place of TMRE or using a chronic treatment of one of the synaptic drugs in place of the mutants and then washing it out before dye treatment. At the very least the caveat of potential drug uptake confounds should be discussed in the manuscript.

As mentioned above in our response to Reviewer #1 we did make a solid effort towards using the ratiometric dye JC-1 to quantify differences in mitochondrial membrane potential in our mutants. In our hands we found this dye to be quite toxic and reduced the incubation time dramatically from previous work. Using JC-1 we saw a reduction in cav1.3 and otof mutants, but this data did not reach significance (p=0.07 for cav1.3 and p=0.34 for otof). This is now shown in Figure 5—figure supplement 3 and is included in our Results section.

Although we did not include this data in the revision, we did try TMRE labeling after a 24 hr treatment with isradipine to block calcium channels over a longer time window. We found that once the isradipine is washed off and the TMRE dye labeling protocol is finished, the membrane potential measurements are normal. This suggests that either blocking calcium channel activity does nothing to TMRE label or it suggests that TMRE label only reflects a snapshot of current activity levels, which may return to a more normal level after drug washout, rather than reflecting any long-term effects of drug treatment. Although these results are really interesting, at this point, significantly more work is needed to distinguishing between these scenarios.

In our Results section, we now highlight the weakness of using single wavelength dyes. We end this explanation by emphasizing why the MitoTimer result is important.

“Our TMRE measurements reflect mitochondrial activity at a single moment in time (Figure 5I-L). In contrast our CellROX Orange and MitoSOX Red measurements likely reflect ROS that have accrued over time (Figure 5A-H). One drawback to using these single wavelength vital dyes is that they rely on comparable dye uptake between our mutants and controls. Although we observed normal Neo-TR entry into hair cells in our mutants, it is possible that vital dye entry is impaired. Therefore, we examined how mitochondrial activity relates to ROS production over time using a genetically encoded indicator of oxidative stress called MitoTimer. Unlike vital dyes, by using a stable transgenic line MitoTimer is present at comparable levels in mutants and controls. MitoTimer localizes to mitochondria and exhibits an oxidation-dependent shift in fluorescence signal from green to red.”